# MindMerger: Efficiently Boosting LLM Reasoning in non-English Languages

**Zixian Huang[1], Wenhao Zhu[1], Gong Cheng[1], Lei Li[2], Fei Yuan[3]***

[1]State Key Laboratory for Novel Software Technology, Nanjing University
[2] Carnegie Mellon University
[3]Shanghai Artificial Intelligence Laboratory
`{zixianhuang, zhuwh}@smail.nju.edu.cn, gcheng@nju.edu.cn`
`leili@cs.cmu.edu, yuanfei@pjlab.org.cn`

## Abstract

Reasoning capabilities are crucial for Large Language Models (LLMs), yet a notable gap exists between English and non-English languages. To bridge this disparity, some works fine-tune LLMs to relearn reasoning capabilities in non-English languages, while others replace non-English inputs with an external model's outputs such as English translation text to circumvent the challenge of LLM understanding non-English. Unfortunately, these methods often underutilize the built-in skilled reasoning and useful language understanding capabilities of LLMs. In order to better utilize the minds of reasoning and language understanding in LLMs, we propose a new method, namely MindMerger, which merges LLMs with the external language understanding capabilities from multilingual models to boost the multilingual reasoning performance. Furthermore, a two-step training scheme is introduced to first train to embeded the external capabilities into LLMs and then train the collaborative utilization of the external capabilities and the built-in capabilities in LLMs. Experiments on three multilingual reasoning datasets and a language understanding dataset demonstrate that MindMerger consistently outperforms all baselines, especially in low-resource languages. Without updating the parameters of LLMs, the average accuracy improved by 6.7% and 8.0% across all languages and low-resource languages on the MGSM dataset, respectively [2].

## 1 Introduction

One of the primary focuses of Artificial Intelligence research currently revolves around improving its reasoning capabilities [Ahn et al., 2024, Minaee et al., 2024], which is derived from the need to enable Large Language Models (LLMs) [Ouyang et al., 2022, Touvron et al., 2023, Jiang et al., 2023a] to think rationally and perform functions like humans [Imani et al., 2023, Jiang et al., 2023b]. Substantial progress has been made in English reasoning [Yu et al., 2023, Yuan et al., 2023a], but the performance in non-English, especially low-resource languages, still lags behind [Shi et al., 2023] due to the scarce of multilingual training data [Touvron et al., 2023].

Existing work tries to use external models to compensate for the deficiencies of LLM in multilingual reasoning. Some works use the relearning-based strategy, which uses translation models to generate multilingual training data for fine-tuning LLMs to relearn reasoning in each language [Chen et al., 2023, Chai et al., 2024]. Some other works use the replacement-based strategy, which use translation models to translate queries from non-English to English text for replacing the non-English input of

---

*Corresponding author.

[2]Our code is available on `https://github.com/CONE-MT/MindMerger`.

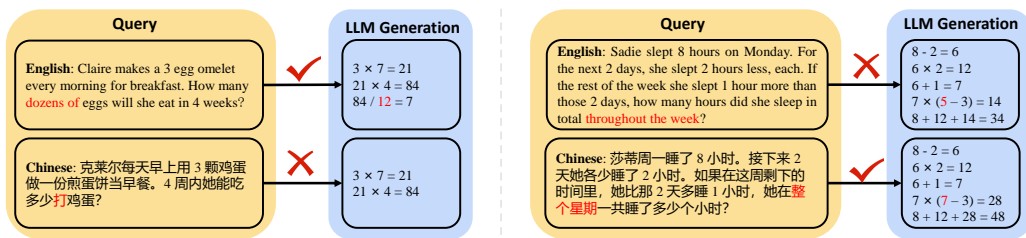

Figure 1: Examples of multilingual mathematical reasoning from the MGSM dataset. LLM can generate correct and incorrect answers when asked in different languages.

LLM [Shi et al., 2023]. Both strategies try to use the translation model to help LLMs master new capabilities, but the insufficient translation quality constrains the performance of these methods.

Moreover, certain capabilities, such as reasoning and language understanding, are built-in in LLMs and should be utilized without the need to develop them from scratch. For the reasoning capabilities, as illustrated in Figure 1, regardless of the language in which the same mathematical question is posed, the correct reasoning process remains consistent. It shows that the reasoning capabilities is universal rather than language-specific [Brannon, 2005]. For language understanding capabilities, while the Chinese question in the first example of Figure 1 may cause a reasoning error by failing in understanding the English-origin term "dozen of", the LLM successfully differentiated between "the week" and "the weekdays" in the Chinese question, contrasting with the failure of the English question in the second example. It shows that although the proficiency may not match that of English, expressions in non-English languages remain valuable to LLMs.

Considering that the built-in reasoning and language understanding capabilities of LLMs need to be better utilized, in this paper, we propose a new method **MindMerger**, which preserves the minds of reasoning and language understanding capabilities in LLMs, and merges the external language understanding capabilities from pre-trained multilingual models to boost the multilingual reasoning effects. To address the challenge of insufficient generation quality of external models, MindMerger feeds the LLM the undecoded query representation from the multilingual model rather than text. Additionally, it uses an augmentation strategy that combines the encoded query representation with the input of LLM to utilize both external and built-in language understanding capabilities.

Given the inconsistency in the representation space, understanding the query representation encoded by multilingual model is not trivial for LLMs. To address this, we propose a two-stage training scheme including the **mapping stage** and the **augmentation stage**. In the mapping stage, we train MindMerger to embed the language capabilities of the multilingual model into the LLM by using accessible general bilingual pairs such as translation data. In the augmentation stage, MindMerger is further trained to collaboratively utilize the built-in capabilities of LLM and the embedded external language capabilities by using query translation task data generated from translation model. Throughout the two stages, only a small number of parameters that connect two models are trained, while both the multilingual model and the LLM are frozen to prevent their built-in capabilities from forgetting.

Extensive experiments are conducted on three multilingual reasoning datasets and a language understanding dataset. Taking on the MGSM dataset [Shi et al., 2023] as an example, MindMerger outperforms all baselines and achieves a lead of at least 6.7% on the average accuracy across all languages, and its performance in low-resource languages is even more significant leading by at least 8.0% (§ 4.3). Compared with the replacement-based method that translates non-English text into English as the LLM input, MindMerger can lead by at least 6.6% in average accuracy based on the same translation model (§ 5.1). Benefiting from the accessible general bilingual pairs used in the mapping stage, the average accuracy across low-resource languages increased by 14.9% (§ 5.2).

Our contributions can be summarized as follows:

- We propose a new method MindMerger to boost the multilingual reasoning of LLMs, which preserves the built-in reasoning and language understanding capabilities of LLMs while augmenting them with the external language understanding capabilities from multilingual models.
- We propose a two-stage training scheme to help MindMerger sequentially learn to embed external capabilities and collaboratively utilize internal and external capabilities.

- MindMerger achieves the best results on four dataset about multilingual reasoning or understanding datasets, notably improving all languages, including low-resource languages, with an average accuracy increase of at least 6.7% and 8.0% on the MGSM datasets.

## 2 Related Work

**Multilingual Reasoning.** There have been some attempts to improve LLM's performance on multilingual reasoning. Several works design crafted prompts to assist LLMs in reasoning [Huang et al., 2023, Qin et al., 2023], but their effectiveness diminishes when used with open-source LLMs like Llama [Touvron et al., 2023] due to limited capacity for multi-step instruction-following [Huang et al., 2023]. Supervised fine-tuning LLMs is another effective way, where some works use translation models to translate query-response [Chen et al., 2023], or query-only [Zhu et al., 2024] to build multilingual task datasets, enabling LLMs to relearn reasoning or language understanding. Some other works utilize external models to generate English text [Shi et al., 2023] to replace the non-English input of LLMs, aiming to utilize the reasoning capabilities of LLMs directly. However, the above methods are limited by the generation quality of the translation model, and the built-in reasoning or language understanding capabilities of LLM are neglected. In contrast, MindMerger avoid the loss introduced by autoregressive decoding and employs an augmentation-based strategy that utilizes the built-in capabilities of LLMs to enhance multilingual reasoning performance.

**Model Merging.** Model Merging is a popular topic in recent LLM research. Generally, it aims to combine an external module to strengthen the capabilities that LLM lacks, such as multi-modal vision capabilities [Liu et al., 2023, Li et al., 2023, Zhu et al., 2023]. Some works [Sun et al., 2021, Bansal et al., 2024] find that interpolating models with multilingual capabilities using cross-attention can improve the performance of multilingual tasks, but research on merging multilingual models and LLMs with English reasoning capabilities is still scarce. Recently, Yoon et al. [2024] merge the encoder of multilingual model and the LLM to improve the multilingual reasoning performance. However, the input of LLMs is completely replaced by the output of a multilingual encoder, making its built-in multilingual capabilities underutilized. In addition, only using English task data for training limits the performance of model merging. Instead of replacing the input of LLMs, we collaboratively utilize the features of the multilingual model and use the available general bilingual pair to train to obtain multilingual reasoning capabilities.

## 3 Approach

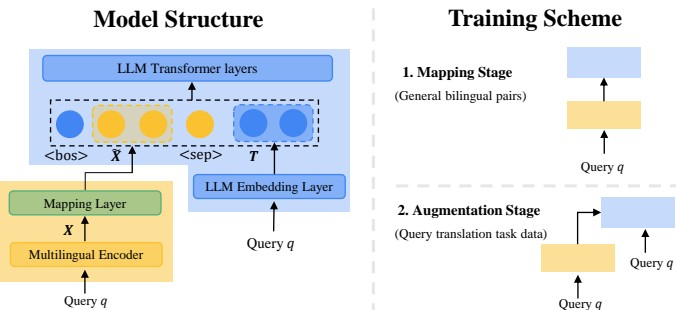

Figure 2: Overview of the model structure and training scheme of MindMerger, which consists of an LLM (blue) and a external model (yellow) and is trained by a two-stage scheme.

Given an LLM with skilled reasoning capabilities and useful language understanding capabilities, our target is to maintain the built-in capabilities and compensate for its shortcomings in non-English language understanding capabilities with an external multilingual model. To this end, as shown in Figure 2, we propose MindMerger that uses the output of the multilingual model as an augmentation complementing to the original input (§ 3.1). We further design a two-stage training scheme to learn the collaborative utilizing both the external and built-in capabilities (§ 3.2).

## 3.1 Model Structure

Given a query $q$ with $l$ tokens, we first utilize the multilingual model to understand and encode it into a representation $\boldsymbol{X}$ that is more general and reduces the challenge of multilingual understanding:

$$\boldsymbol{X} = \texttt{Encoder}(q), \tag{1}$$

where $\texttt{Encoder}(\cdot)$ is a pre-trained multilingual model, typically using its encoder part, and $\boldsymbol{X} \in \mathbb{R}^{l \times d_1}$ is the hidden state output of the multilingual model with a dimension of $d_1$.

The representation $\boldsymbol{X}$ resides in the multilingual model space, which is separate from the LLM space and cannot be used directly. Therefore, we introduce a mapping layer:

$$\widetilde{\boldsymbol{X}} = \texttt{Mapping}(\boldsymbol{X}), \tag{2}$$

where $\widetilde{\boldsymbol{X}} \in \mathbb{R}^{l \times d_2}$ is the projection of $\boldsymbol{X}$ on the space of LLM. Unless otherwise stated, the implementation of $\texttt{Mapping}(\cdot)$ is a two-layer multi-layer perceptron (MLP).

The acquisition of $\boldsymbol{X}$ utilizes the language understanding capabilities of the external multilingual model. In order to take advantage of the built-in capabilities of the LLM, we prompt it directly:

$$\boldsymbol{T} = \texttt{Embedding}(q), \tag{3}$$

where $\boldsymbol{T} \in \mathbb{R}^{l \times d_2}$ is the representation of query $q$ in the space of LLM and $\texttt{Embedding}(\cdot)$ is the embedding layer of LLM.

Then, we concatenate the query representation from the multilingual model and the LLM for the collaborative utilizing of LLM's capabilities:

$$(\widetilde{\boldsymbol{X}}, \boldsymbol{T}) = [\langle \texttt{bos} \rangle; \widetilde{\boldsymbol{X}}; \langle \texttt{sep} \rangle; \boldsymbol{T}], \tag{4}$$

where $\langle \texttt{bos} \rangle \in \mathbb{R}^{d_2}$ is the representation of the start token of LLM and $\langle \texttt{sep} \rangle \in \mathbb{R}^{d_2}$ is a trainable variable that denotes the boundary of $\widetilde{\boldsymbol{X}}$. Finally, $(\widetilde{\boldsymbol{X}}, \boldsymbol{T})$ is fed to LLM to generate the response.

## 3.2 Two-Stage Training

The training scheme of MindMerger is divided into two stages: mapping stage and augmentation stage. The former helps LLM learn to use the capabilities of multilingual model, and the latter helps LLM collaboratively utilize its own and the capabilities from multilingual model. Examples of training data for each stage are shown in Appendix D.

**Mapping Stage.** Given the representation spaces between multilingual model and LLM are distant different, it is not trivial for LLM to understand and utilize the external capabilities provided by multilingual model. In order to better learn to utilize external capabilities, we force MindMerger to focus on input from the multilingual model using the replacement-based strategy during this stage. Specifically, we simplify the input of Equation (4) as follows:

$$\widetilde{\boldsymbol{M}} = [\langle \texttt{bos} \rangle; \widetilde{\boldsymbol{X}}; \langle \texttt{sep} \rangle]. \tag{5}$$

Since general bilingual pairs such as translation data is in vast availability, we use it from various languages to English to train MindMerger in this stage. The loss function is outlined as follows:

$$- \arg \min_{\sigma} \log \mathcal{P}(Y | \widetilde{\boldsymbol{M}}, \theta, \phi, \sigma), \tag{6}$$

where $Y$ is the text of training target, $\theta$ and $\phi$ are the parameters of the multilingual model and LLM respectively, which are frozen during the training to prevent forgetting, and $\sigma$ contains the trainable parameters of the mapping layer in Equation (2) and the boundary token $\langle \texttt{sep} \rangle$.

**Augmentation Stage.** Although MindMerger have learned to utilize external capabilities in the mapping stage, the replacement-based strategy may cause LLM to neglect the use of its own built-in capabilities. To help MindMerger further learn to merge capabilities from external and built-in LLM, in this stage we use the augmentation-based strategy as described in the Equation (4). The loss function is calculated as follows:

$$- \arg \min_{\hat{\sigma}} \log \mathcal{P}(Y | (\widetilde{\boldsymbol{X}}, \boldsymbol{T}), \theta, \phi, \hat{\sigma}), \tag{7}$$

where $\hat{\sigma}$ is initiated from the checkpoint of $\sigma$ trained at the mapping stage, and the parameters of the multilingual model and LLM represented as $\theta$ and $\phi$, respectively, are kept constant during training. In this stage, the query translation task data generated from public translation models is used as the training data to adapt MindMerger to downstream task.

# 4 Experiments

## 4.1 Compared Methods

**Our Methods.** Two variants of MindMerger were implemented. (1) The implementation described in § 3.1 is denoted as **MindMerger-Soft**. (2) **MindMerger-Hard** augments the prompts of LLM with the translated query given by the translation model (Appendix D for the details of prompts).

**Baselines.** We compared our methods with three categories of baselines. (1) One basic method **MonoReason** [Yu et al., 2023, Zhu et al., 2024] which is fine-tuned on the English task dataset. (2) Three relearning-based methods that use task data with query translation, including the full-parameter fine-tuning model **MultiReason-Full** [Zhu et al., 2024], parameter-efficient fine-tuning model **MultiReason-Lora** [Hu et al., 2022], and the state-of-the-art method **QAlign** [Zhu et al., 2024] which first learns language understanding by training LLM on query translation data and then further fine-tunes LLM as MonoReason. (3) Two replacement-based methods that introduce external models, including **Translate-En** [Shi et al., 2023] and **LangBridge** [Yoon et al., 2024]. Translate-En is a hard replacement-based which translates the query into English to replace the prompt of LLM. LangBridge is a soft replacement-based method which replaces the input of LLMs with the hidden states output by mT5-xl [Xue et al., 2021]. The prompts of each baselines are presented in Appendix D.

**Details.** Unless otherwise stated, we used the encoder part of mT5-xl [Xue et al., 2021] as the multilingual model in our methods, used the NLLB200-3.3B as the translation model for baselines, and used the Llama 2-7B as the LLM for all methods. The influence of different multilingual models including encoder-only, decoder-only and encoder-decoder architectures will be analyzed in § 5.1. Both MindMerger and all the baselines, except QAlign, are based on the same MonoReason model. Additionally, MonoReason and QAlign are trained based on the same LLM. Specifically, we used the publicly available checkpoint given by Yu et al. [2023] as MonoReason model for mathematical reasoning task. For all models, we set learning rate=2e-5, batch size=128, max length=512, and epoch=3 and used 8 NVIDIA A100 GPUs for training.

## 4.2 Datasets

**Evaluation Datasets.** We experimented MindMerger with the latest multilingual mathematical reasoning **MGSM** [Shi et al., 2023] and **MSVAMP** [Chen et al., 2023], where MSVAMP serves as an out-of-domain test set. In order to evaluate the diverse multilingual reasoning capabilities of the models, a challenging multilingual dataset **X-CSQA** [Lin et al., 2021] in commonsense reasoning task and a language understanding dataset **XNLI** [Conneau et al., 2018] in natural language inference (NLI) task were used. The statistics of these datasets are presented in Appendix D.

**Training Datasets.** Three categories of training data were used in our methods and baselines. (1) **General bilingual pairs**. We used the translation data from the multilingual language to English and randomly sampled 100K of data for each language (except English) from the Lego-MT [Yuan et al., 2023b] dataset, which is a large-scale translation dateset that contains all the languages that involved in our experiments. (2) **English task data**. We used MetaMathQA [Yu et al., 2023] and MultiNLI [Williams et al., 2018] datasets for mathematical reasoning and NLI task, respectively. Similar to Huang et al. [2022], we unified the training set of X-CSQA, OpenBookQA [Mihaylov et al., 2018], ARC [Bhakthavatsalam et al., 2021] and QASC [Khot et al., 2020] to train commonsense reasoning task more fully. (3) **Query translation task data**. We used the translated results given by Chen et al. [2023] and the official dev set of XNLI for mathematical reasoning and the NLI task, respectively, and translated the X-CSQA training set based on M2M100-1.2B [Fan et al., 2021].

## 4.3 Experimental Results

**MindMerger improves LLM performance on all datasets, especially benefiting low-resource languages.** MindMerger-Soft has an average accuracy better than all other baselines at least 6.7%, 3.2% on the MGSM and MSVAMP datasets in Table 1, demonstrating its remarkable multilingual reasoning capabilities. For the commonsense reasoning task in Table 2, MindMerger-Soft also significantly leads all baselines by at least 4.1%. In Table 3, MindMerger-Soft demonstrates advantages in language understanding, which significantly outperformed all the baselines (with p-value < 0.01). MindMerger-Hard achieves the best results except MindMerger-Soft on three out of four datasets, demonstrating the advantages of augmentation-based methods over other categories of methods.

Table 1: Experimental results on MGSM and MSVAMP datasets. Lrl., Hrl., and Avg. represent the average accuracy across low-resource languages, high-resource languages, and all languages, respectively. Referring to Shi et al. [2023], we regard Bn, Th, and Sw as low-resourse languages, and regard the remaining languages as high-resource languages. We used the checkpoints given by Yu et al. [2023] as the MonoReason models.

| MGSM | Bn | Th | Sw | Ja | Zh | De | Fr | Ru | Es | En | Lrl. | Hrl. | Avg. |
|---|---|---|---|---|---|---|---|---|---|---|---|---|---|
| MonoReason [Yu et al., 2023, Zhu et al., 2024] | 6.8 | 7.2 | 6.8 | 36.4 | 38.4 | 55.2 | 54.4 | 52.0 | 57.2 | 68.8 | 6.9 | 51.8 | 38.3 |
| MultiReason-Lora [Hu et al., 2022] | 29.6 | 35.2 | 28.0 | 52.0 | **54.8** | 59.6 | **58.4** | 62.4 | 59.6 | 64.8 | 30.9 | 58.8 | 50.4 |
| MultiReason-Full [Zhu et al., 2024] | 33.2 | 40.0 | 42.0 | 42.0 | 42.0 | 45.2 | 44.8 | 45.2 | 48.0 | 52.0 | 38.4 | 45.6 | 43.4 |
| QAlign [Zhu et al., 2024] | 32.4 | 39.6 | 40.4 | 44.0 | 48.4 | 54.8 | 56.8 | 52.4 | 59.6 | 68.0 | 37.5 | 54.9 | 49.6 |
| LangBridge [Yoon et al., 2024] | 42.8 | 50.4 | 43.2 | 40.0 | 45.2 | 56.4 | 50.8 | 52.4 | 58.0 | 63.2 | 45.5 | 52.3 | 50.2 |
| Translate-En [Shi et al., 2023] | 48.4 | 37.6 | 37.6 | 49.2 | 46.8 | 60.4 | 56.4 | 47.6 | 59.6 | 65.5 | 41.2 | 55.1 | 50.6 |
| MindMerger-Hard | 46.0 | 36.0 | 48.4 | 52.4 | 54.4 | 60.4 | 56.0 | 60.4 | **62.0** | 71.2 | 43.5 | 59.5 | 54.7 |
| MindMerger-Soft | **50.4** | **52.8** | **57.2** | **54.4** | 53.6 | **61.2** | 57.6 | 60.8 | 58.4 | 66.8 | **53.5** | **59.0** | **57.3** |

| MSVAMP | Bn | Th | Sw | Ja | Zh | De | Fr | Ru | Es | En | Lrl. | Hrl. | Avg. |
|---|---|---|---|---|---|---|---|---|---|---|---|---|---|
| MonoReason [Yu et al., 2023, Zhu et al., 2024] | 12.5 | 15.8 | 17.2 | 54.0 | 55.9 | 63.9 | 65.2 | 58.1 | 64.3 | 67.1 | 15.2 | 61.2 | 47.4 |
| MultiReason-Lora [Hu et al., 2022] | 39.4 | 43.8 | 39.1 | 55.2 | 55.1 | 60.4 | 59.1 | 56.8 | 60.6 | 64.2 | 40.8 | 58.8 | 53.4 |
| MultiReason-Full [Zhu et al., 2024] | 34.8 | 38.1 | 39.8 | 43.4 | 42.9 | 45.6 | 45.8 | 45.0 | 46.1 | 46.8 | 37.6 | 45.1 | 42.8 |
| QAlign [Zhu et al., 2024] | 41.7 | 47.7 | **54.8** | 58.0 | 55.7 | 62.8 | 63.2 | 61.1 | 63.3 | 65.3 | 48.1 | 61.3 | 57.2 |
| LangBridge [Yoon et al., 2024] | 46.8 | 46.3 | 42.1 | 45.5 | 50.4 | 58.1 | 57.0 | 55.8 | 60.6 | 60.6 | 45.1 | 54.9 | 50.9 |
| Translate-En [Shi et al., 2023] | 47.9 | 51.3 | 43.1 | 50.4 | 55.8 | 43.9 | 50.9 | 53.4 | 51.4 | 60.6 | 47.4 | 52.3 | 50.9 |
| MindMerger-Hard | 43.6 | 46.7 | 50.6 | **61.4** | 60.6 | **65.9** | **65.6** | 62.1 | 65.0 | 67.8 | 47.0 | **64.1** | 58.9 |
| MindMerger-Soft | **52.0** | **53.4** | 54.0 | 59.0 | **61.7** | 64.1 | 64.0 | **63.3** | 65.0 | 67.7 | **53.1** | 63.5 | **60.4** |

MindMerger-Soft stands out notably in enhancing the reasoning capabilities of low-resource languages. Compared to all baselines, it delivers an average accuracy improvement of at least 8.0% and 5.0% across low-resource languages on the MGSM, MSVAMP datasets.

**Utilizing the built-in reasoning capabilities of the LLM rather than relearning for non-English is a more effective way to boost multilingual reasoning capabilities.** It is not easy for relearning-based methods to improve the performance of high-resource and low-resource languages simultaneously. For instance, MultiReason-Full improves its performance in low-resource languages from an average of 6.9% to 38.4% on the MGSM dataset, but its performance in high-resource languages declines from an average of 51.8% to 45.6%. Compared with the three relearning-based baselines, MindMerger-Soft achieves a lead on both low-resource and high-resource languages and outperforms them by at least 6.9% on the MGSM datasets for the average accuracy across all languages.

**Utilizing the built-in language understanding capabilities of LLM rather than replacing them boosts multilingual reasoning capabilities more significantly.** Comparing Translate-En and MindMerger-Hard which are completely consistent except for the input strategy, the augmentation-based MindMerger-Hard has a higher average accuracy than the replacement-based Translate-En of 4.4% and 11.8% across high-resource languages in the MGSM and MSVAMP datasets respectively, showing that non-English input can help LLM enhance query understanding. Although the understanding capabilities of LLM in low-resource languages are limited, MindMerger-Hard still leads Translate-En by 2.3% on the MSGM dataset. Larger boost comes from MindMerger-Soft, which leads all replacement-based methods with an average accuracy of at least 6.7% across all languages.

Table 2: Results on the X-CSQA dataset. Avg. represents the average accuracy across all languages.

| X-CSQA | Sw | Ur | Hi | Ar | Vi | Ja | Pl | Zh | Nl | Ru | It | De | Pt | Fr | Es | En | Avg. |
|---|---|---|---|---|---|---|---|---|---|---|---|---|---|---|---|---|---|
| MonoReason [Yu et al., 2023, Zhu et al., 2024] | 24.2 | 25.1 | 32.9 | 32.3 | 50.9 | 49.1 | 50.6 | 56.5 | 57.5 | 56.0 | 56.0 | 61.2 | 61.7 | 63.5 | 64.0 | 76.3 | 51.3 |
| MultiReason-Lora [Hu et al., 2022] | 25.1 | 32.0 | 39.2 | 42.2 | 56.6 | **55.9** | 60.6 | 62.2 | 61.3 | 62.8 | 66.3 | 64.9 | 66.2 | 67.4 | 67.7 | **79.3** | 56.9 |
| MultiReason-Full [Zhu et al., 2024] | 27.6 | 29.2 | 32.0 | 28.7 | 38.8 | 38.7 | 45.5 | 43.8 | 45.9 | 46.5 | 50.2 | 49.1 | 51.2 | 52.1 | 54.3 | 67.2 | 43.8 |
| QAlign [Zhu et al., 2024] | 35.1 | 32.6 | 37.8 | 36.3 | 50.5 | 49.2 | 51.3 | 54.8 | 56.3 | 58.3 | 58.8 | 59.8 | 60.3 | 63.1 | 63.1 | 75.7 | 52.3 |
| LangBridge [Yoon et al., 2024] | 31.8 | 30.5 | 30.6 | 30.6 | 33.3 | 33.9 | 39.8 | 39.8 | 38.4 | 35.1 | 39.1 | 37.4 | 36.3 | 38.2 | 38.4 | 44.4 | 36.1 |
| Translate-En [Shi et al., 2023] | 36.5 | 41.3 | **48.4** | 44.6 | 51.8 | 47.1 | 53.3 | 51.5 | 55.0 | 54.4 | 56.3 | 57.3 | 54.7 | 57.2 | 55.5 | 71.3 | 52.3 |
| MindMerger-Hard | 33.1 | 29.9 | 40.4 | 37.7 | 52.9 | 49.9 | 54.7 | 55.4 | 58.0 | 58.0 | 59.7 | 58.6 | 61.9 | 62.5 | 63.6 | 75.2 | 53.1 |
| MindMerger-Soft | **45.5** | **46.2** | **48.4** | **51.4** | **60.6** | 53.9 | **63.3** | **62.9** | **63.8** | **63.7** | **66.8** | **67.0** | **67.1** | **68.1** | **69.1** | 78.1 | **61.0** |

# 5 Analysis

## 5.1 The Usage of Multilingual Model

**MindMerger can flexibly interpolate among various multilingual models.** We experimented with a wide variety of multilingual models, including decoder-only models mGPT [Shliazhko et al., 2022], encoder-only models mBERT [Devlin et al., 2018] and XLM-RoBERTa-large [Conneau et al., 2019],

Table 3: Results on the XNLI dataset. Avg. represents the average accuracy across all languages.

| XNLI | Sw | Ur | Hi | Th | Ar | Tr | El | Vi | Zh | Ru | Bg | De | Fr | Es | En | Avg. |
|---|---|---|---|---|---|---|---|---|---|---|---|---|---|---|---|---|
| MonoReason [Yu et al., 2023, Zhu et al., 2024] | 45.9 | 49.2 | 55.7 | 55.4 | 60.9 | 61.9 | 63.7 | 73.7 | 74.7 | 77.6 | 76.7 | 80.6 | 82.2 | 82.8 | **90.0** | 68.7 |
| MultiReason-Lora Hu et al. [2022] | 45.9 | 49.3 | 56.4 | 55.7 | 60.9 | 61.9 | 64.7 | 73.7 | 74.7 | 76.7 | 76.7 | 80.6 | 82.2 | 82.8 | **90.0** | 68.9 |
| MultiReason-Full [Zhu et al., 2024] | 56.3 | 57.5 | 61.7 | 60.1 | 61.7 | 65.6 | 67.0 | 73.7 | 79.1 | 79.7 | 78.7 | 82.3 | 82.9 | 83.9 | 88.8 | 71.9 |
| QAlign [Zhu et al., 2024] | 65.2 | 62.2 | 63.3 | 65.2 | 67.0 | 67.9 | 66.5 | 73.7 | 76.6 | 79.2 | 79.4 | 80.9 | 83.1 | 83.8 | 89.1 | 73.5 |
| LangBridge [Yoon et al., 2024] | **71.7** | 66.9 | 71.1 | **72.4** | 75.2 | 74.8 | 79.1 | 78.5 | 77.4 | 77.4 | 79.6 | 78.8 | 79.9 | 80.5 | 83.4 | 76.5 |
| Translate-En [Shi et al., 2023] | 65.3 | 61.6 | 68.7 | 69.5 | 68.9 | 74.5 | **79.3** | 76.7 | 74.8 | 76.0 | 80.8 | 80.6 | 80.4 | 81.4 | 87.4 | 75.1 |
| MindMerger-Hard | 65.7 | 56.4 | 58.3 | 64.1 | 63.6 | 70.0 | 62.2 | 56.6 | 61.8 | 58.1 | 61.7 | 64.2 | 61.2 | 63.3 | 80.8 | 63.2 |
| MindMerger-Soft | 66.6 | **69.4** | **74.7** | 71.8 | **76.2** | **75.7** | 78.5 | **80.3** | **80.0** | 80.7 | **82.4** | **83.5** | **83.9** | **84.4** | 88.7 | **78.4** |

Table 4: Merging with different multilingual models on the MGSM dataset. # Parm represents the number of parameters of the used external model. Lrl., Hrl., and Avg. represent the average accuracy across low-resource languages, high-resource languages, and all languages, respectively. Referring to Shi et al. [2023], we regard Bn, Th, and Sw as low-resourse languages, and regard the remaining languages as high-resource languages.

| MGSM | # Parm | Bn | Th | Sw | Ja | Zh | De | Fr | Ru | Es | En | Lrl. | Hrl. | Avg. |
|---|---|---|---|---|---|---|---|---|---|---|---|---|---|---|
| Translate-En | | | | | | | | | | | | | | |
| M2M100-418M | 484 M | 30.0 | **38.0** | 38.8 | 31.6 | **50.8** | 52.0 | 50.0 | 42.4 | 54.0 | **65.5** | 35.6 | 49.5 | 44.7 |
| M2M100-1.2B | 1,239 M | 42.4 | 34.0 | **49.6** | 40.8 | 42.8 | 55.2 | 50.4 | **49.2** | 46.8 | **65.5** | **42.0** | 50.1 | 47.1 |
| NLLB-200-1.3B | 1,371 M | 46.0 | 32.0 | 40.4 | 47.2 | 45.6 | 55.2 | 51.2 | 46.0 | 55.2 | **65.5** | 39.5 | 52.3 | 47.9 |
| NLLB-200-3.3B | 3,345 M | **48.4** | 37.6 | 37.6 | **49.2** | 46.8 | **60.4** | 56.4 | 47.6 | **59.6** | **65.5** | 41.2 | **55.1** | 50.6 |
| MindMerger-Hard | | | | | | | | | | | | | | |
| M2M100-418M | 484 M | 39.6 | 28.0 | 36.4 | 49.2 | 48.8 | 60.0 | 56.4 | 55.6 | 58.4 | 64.8 | 34.7 | 56.2 | 49.7 |
| M2M100-1.2B | 1,239 M | 40.0 | **36.0** | 47.6 | **52.4** | 50.8 | 58.0 | 56.4 | 60.8 | 61.2 | 66.8 | 41.2 | 58.1 | 53.0 |
| NLLB-200-1.3B | 1,371 M | 44.0 | 30.0 | 42.8 | 48.0 | 53.6 | **61.6** | 56.4 | 54.8 | **63.6** | 70.8 | 38.9 | 58.4 | 52.6 |
| NLLB-200-3.3B | 3,345 M | 46.0 | 36.0 | 48.4 | 52.4 | 54.4 | 60.4 | 56.0 | 60.4 | 62.0 | **71.2** | 43.5 | 59.5 | 54.7 |
| MindMerger-Soft | | | | | | | | | | | | | | |
| mGPT | 1,418 M | 19.6 | 20.4 | 15.6 | 42.8 | 48.0 | 59.2 | 59.6 | 54.0 | **61.2** | 64.0 | 18.5 | 55.5 | 44.4 |
| mBERT | 178 M | 30.8 | 37.6 | 46.8 | 50.0 | 48.8 | 55.6 | 52.4 | 59.6 | 60.8 | 66.4 | 38.4 | 56.2 | 50.9 |
| XLM-RoBERTa-large | 560 M | 44.0 | 52.4 | 50.4 | 52.4 | 54.0 | 60.8 | 58.4 | 56.8 | 56.8 | 66.4 | 48.9 | 57.9 | 55.2 |
| M2M100-418M | 282 M | 49.2 | **52.8** | 46.0 | 48.8 | 52.4 | 59.6 | 58.0 | 59.2 | 60.8 | 65.6 | 49.3 | 57.8 | 55.2 |
| M2M100-1.2B | 635 M | 49.6 | 52.4 | 53.2 | 52.8 | **54.4** | 60.0 | 56.4 | 60.0 | 58.0 | 66.0 | 51.7 | 58.2 | 56.3 |
| NLLB-200-1.3B | 766 M | 45.6 | 47.6 | **57.6** | 54.4 | 52.4 | 57.2 | 57.2 | **60.8** | 60.8 | 66.8 | 50.3 | 58.5 | 56.2 |
| NLLB-200-3.3B | 1,733 M | **52.4** | 51.6 | 53.6 | 52.8 | 53.2 | 60.4 | **60.0** | 60.4 | 60.4 | **67.6** | 52.5 | **59.3** | 57.2 |
| mT5-large | 564 M | 40.4 | 47.2 | 53.6 | 47.6 | 51.6 | 59.2 | 55.2 | 57.6 | 56.8 | 66.4 | 47.1 | 56.3 | 53.6 |
| mT5-xl | 1,670 M | 50.4 | **52.8** | 57.2 | **54.4** | 53.6 | **61.2** | 57.6 | **60.8** | 58.4 | 66.8 | **53.5** | 59.0 | **57.3** |

and the encoder part of encoder-decoder models M2M100 [Fan et al., 2021], NLLB-200 [Costa-jussà et al., 2022] and mT5 [Xue et al., 2021]. The experimental results in Table 4 show that the encoder part of the encoder-decoder model and the encoder-only model are more suitable interpolate into MindMerger-Soft than the decoder-only model, which can achieve better performance than mGPT with a smaller number of parameters. M2M100 is the most cost-effective model, reaching or even exceeding the performance of XLM-RoBERTa-large and mT5-large while using only half of the parameters of XLM-RoBERTa-large and mT5-large.

**Our augmentation-based strategy outperforms the translate-then-replace strategy.** Comparing Translate-En and MindMerger-Hard which only differ in input strategy, augmentation-based MindMerger-Hard, based on the same translation model, consistently exceeds Translate-En by 5.0%, 5.9%, 4.7%, and 4.1% in the average accuracy. MindMerger-Soft further expands its lead with an increment of the average accuracy by at least 6.6% based on the same translation model.

**MindMerger-Soft is a better utilization of existing multilingual model.** As shown in Table 4, the performance of MindMerger-Soft consistently exceeds MindMerger-Hard under the same multilingual model with increases of average accuracy of 5.5% and 3.3% on two versions of M2M100, and 3.6% and 2.5% on two versions of NLLB-200. Although only the encoder part of the multilingual model is used, MindMerger-Soft merges LLM with a dense representation rather than decoded text based on sparse bag-of-words, enhancing the effectiveness of utilizing multilingual model.

**A more powerful multilingual model can better enhance the multilingual capabilities.** We compared the different sizes of each encoder-decoder models and consistently observed that the larger version outperformed the smaller one in Table 4. With the help of the larger model size, the average accuracy is improved to 1.1%, 1.0%, and 3.7% on M2M100, NLLB-200, and mT5, respectively. The improvement in low-resource languages is even more obvious with an average increase in accuracy of 1.1%, 1.0% and 3.7% in M2M100, NLLB-200, and mT5, respectively. This underscores the greater imperative to enhance language understanding in low-resource languages.

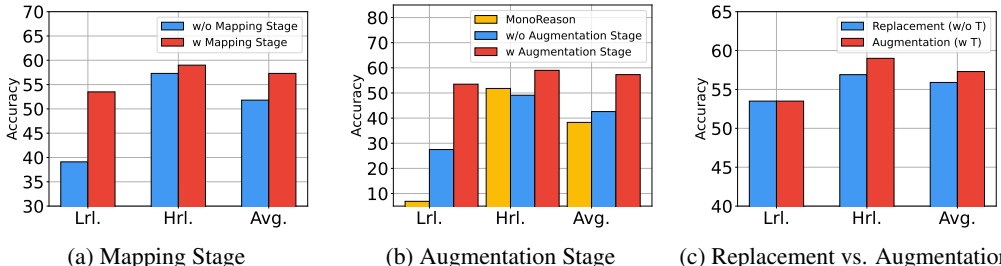

|  | (a) Mapping Stage | (b) Augmentation Stage | (c) Replacement vs. Augmentation |

Figure 3: Ablation experiments of MindMerger-Soft on the MGSM dataset. Lrl., Hrl., and Avg. represent the average accuracy across low-resource languages, high-resource languages, and all languages, respectively. Referring to Shi et al. [2023], we regard Bn, Th, and Sw as low-resourse languages, and regard the remaining languages as high-resource languages.

## 5.2 Ablation Studies

**Mapping Stage.** In Figure 3a, we ablated the mapping stage to observe its necessity. A significant drop in the performance of low-resource languages can be observed when ablating the mapping stage, which shows that the accessible general bilingual pairs are beneficial to help MindMerger-Soft understand the low-resource language information that exists in the multilingual model representation space. More detailed are reported in Appendix B.1.

**Augmentation Stage.** In Figure 3b, we ablated the augmentation stage to observe its necessity. It can be observed that after ablating the augmentation stage, even without using any task-related data, MindMerger-Soft can still outperform MonoReason on low-resource languages, which demonstrates the generalization of MindMerger-Soft that benefits from using accessible general bilingual pairs in the mapping stage. Furthermore, when the augmentation stage is added, MindMerger-Soft exhibits a significant improvement, demonstrating its effectiveness in learning the utilization of both external and built-in capabilities. More detailed are reported in Appendix B.2.

**Replacement vs. Augmentation.** MindMerger uses the representation $X$ outputted by multilingual model to augment the LLM's original representation $T$ rather than replace it to better utilize the built-in capabilities of LLMs. To verify the advantages of the augmentation-based strategy, we removed $T$ to make MindMerger-Soft a replacement-based method. As shown in Figure 3c. although with exactly the same training data and process, the performance of the replacement-based MindMerger-Soft drops significantly, indicating that it is valuable to use the built-in capabilities of LLM to understand the original input. More detailed are reported in Appendix B.3.

Table 5: Results on the MGSM dataset based on MetaMath-Llama-13B and MetaMath-Mistral-7B. Lrl., Hrl., and Avg. represent the average accuracy across low-resource languages, high-resource languages, and all languages, respectively. Referring to Shi et al. [2023], we regard Bn, Th, and Sw as low-resourse languages, and regard the remaining languages as high-resource languages.

| MetaMath-Llama-13B | Bn | Th | Sw | Ja | Zh | De | Fr | Ru | Es | En | Lrl. | Hrl. | Avg. |
|---|---|---|---|---|---|---|---|---|---|---|---|---|---|
| MonoReason [Yu et al., 2023, Zhu et al., 2024] | 12.0 | 8.8 | 6.4 | 48.0 | 56.0 | 64.0 | 63.6 | 62.0 | 67.2 | **70.8** | 9.1 | 61.7 | 45.9 |
| MultiReason-Lora [Hu et al., 2022] | 44.0 | 49.2 | 40.8 | 58.0 | 61.2 | 64.0 | 64.4 | 64.8 | 67.6 | 68.4 | 44.7 | 64.1 | 58.2 |
| MultiReason-Full [Zhu et al., 2024] | 44.8 | 51.6 | 50.8 | 58.0 | 61.6 | 64.8 | 59.2 | 60.8 | 67.6 | 66.4 | 49.1 | 62.6 | 58.6 |
| QAlign [Zhu et al., 2024] | 38.4 | 49.6 | 46.0 | 52.4 | 59.2 | 62.0 | 62.4 | 64.4 | 67.2 | 69.2 | 44.7 | 62.4 | 57.1 |
| LangBridge [Yoon et al., 2024] | 39.2 | 42.8 | 42.0 | 33.6 | 42.0 | 55.2 | 54.8 | 58.8 | 60.8 | 65.2 | 41.3 | 52.9 | 49.4 |
| Translate-En [Shi et al., 2023] | 34.8 | 54.0 | 44.4 | 44.4 | 58.0 | 53.6 | 54.0 | 45.6 | 62.4 | **70.8** | 44.4 | 55.5 | 52.2 |
| MindMerger-Hard | 48.0 | 38.4 | 53.6 | 51.6 | 52.8 | **66.8** | 61.6 | 60.8 | 68.4 | 67.6 | 46.7 | 61.4 | 57.0 |
| MindMerger-Soft | **55.2** | **59.6** | **56.4** | **60.0** | 60.4 | 65.2 | **63.6** | **68.0** | 69.6 | 68.8 | **57.1** | **65.1** | **62.7** |
| **MetaMath-Mistral-7B** | **Bn** | **Th** | **Sw** | **Ja** | **Zh** | **De** | **Fr** | **Ru** | **Es** | **En** | **Lrl.** | **Hrl.** | **Avg.** |
| MonoReason [Yu et al., 2023, Zhu et al., 2024] | 38.4 | 34.8 | 16.8 | 50.8 | 57.2 | 70.4 | 70.8 | 67.2 | 71.2 | 78.0 | 30.0 | 66.5 | 55.6 |
| MultiReason-Lora [Hu et al., 2022] | 46.8 | 51.2 | 39.6 | 54.4 | 62.4 | **72.0** | 66.0 | 68.4 | 70.0 | 76.0 | 45.9 | 67.0 | 60.7 |
| MultiReason-Full [Zhu et al., 2024] | 18.4 | 26.4 | 26.8 | 30.8 | 28.8 | 32.4 | 34.8 | 32.0 | 38.0 | 39.6 | 23.9 | 33.8 | 30.8 |
| QAlign [Zhu et al., 2024] | 45.6 | 51.2 | 55.2 | 49.4 | 57.2 | 59.2 | 59.8 | 60.2 | 63.6 | 65.8 | 50.7 | 59.3 | 56.7 |
| LangBridge [Yoon et al., 2024] | 50.0 | **60.0** | 47.2 | 58.4 | 65.6 | 68.4 | 68.8 | 68.4 | 65.6 | 65.6 | 52.4 | 65.8 | 61.8 |
| Translate-En [Shi et al., 2023] | 54.6 | 58.7 | 47.7 | 57.2 | 63.1 | 50.4 | 56.7 | 64.9 | 58.6 | 69.7 | 53.7 | 60.1 | 58.2 |
| MindMerger-Hard | 52.4 | 48.4 | **57.6** | **62.4** | 60.0 | 66.4 | 66.8 | **69.6** | **71.6** | 76.4 | 52.8 | 67.6 | 63.2 |
| MindMerger-Soft | **57.6** | 59.6 | 53.2 | 57.2 | **68.8** | 69.2 | **69.6** | 68.4 | **71.6** | **79.2** | **56.8** | **69.1** | **65.4** |

## 5.3 Merging with Different LLMs

MindMerger can be flexibly integrated with different LLMs. To verify this, we conducted experiments on a different type of LLM, MetaMath-Mistral-7B [Jiang et al., 2023a, Yu et al., 2023], and a larger size, MetaMath-Llama-13B [Touvron et al., 2023, Yu et al., 2023]. The experimental results are shown in Table 5, MindMerger-Soft has achieved superior performance across various baselines. The average accuracy of MindMerger-Soft on the MetaMath-Llama-13B and MetaMath-Mistral-7B versions is at least 4.1% and 4.7% higher than all baselines, respectively, demonstrating the potential of extending MindMerger to a wider range of LLMs.

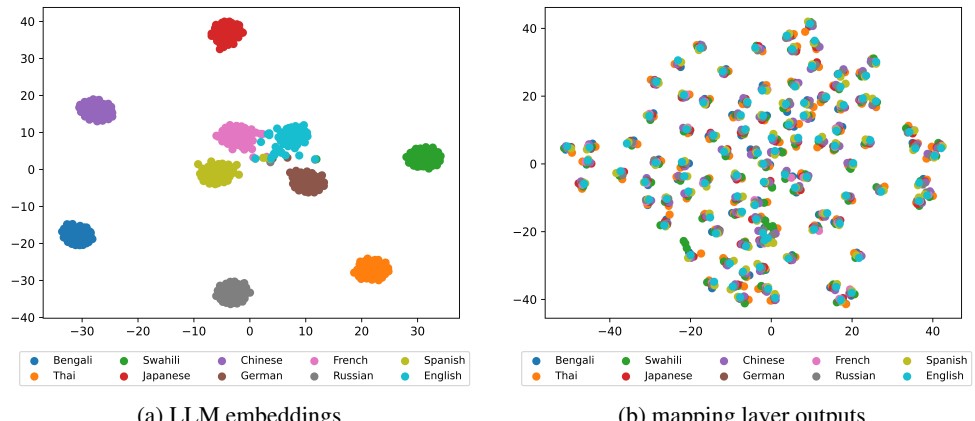

(a) LLM embeddings          (b) mapping layer outputs

Figure 4: T-SNE visualization in the spaces of the LLM embeddings and mapping layer outputs.

## 5.4 Representation Space Changes

For each language, we selected the same 100 texts from the Flores-101 [Fan et al., 2021] dataset, used the mean pooling operation to obtain the representation vectors of the LLM embedding and the hidden states output by and the mapping layer, and visualized it based on T-SNE [Van der Maaten and Hinton, 2008]. As shown in Figure 4, the representation spaces of non-English on the LLM embedding are independent and away from English, especially in low-resource languages, which leads to understanding challenges for non-English and the inability to use built-in reasoning capabilities. By contrast, as shown in Figure 4b, the representations of all languages outputted by the mapping layer almost overlap with English, which reduces the difficulty for LLM to understand non-English languages and enables various languages to utilize the built-in reasoning capabilities.

## 5.5 Supplementary Experiments

We experimented with several supplementary experiments, including the influence of training dataset size used in augmentation stage (Appendix A.1), the selection of mapping layers structure (Appendix A.2), the usage of encoder-decoder model in MindMerger-Soft (Appendix A.3), the quantitative analysis on representation space changes (Appendix A.4), and the translation performance of MindMerger-Soft after mapping stage (Appendix A.5).

## 6 Conclusion

This paper explores a way to more fully utilize the built-in capabilities of LLMs to improve multilingual reasoning effects. We proposed MindMerger to merge the expert multilingual capabilities in the multilingual model with the skilled reasoning and not very proficient but useful multilingual capabilities in the LLM. Through more fully utilizing the potential of LLM and more effective fusion of multilingual models, the performance of MindMerger exceeds all baselines on three reasoning datasets and a language understanding dataset. In the future, we will explore the possibility of MindMerger empowering more professional skills besides reasoning, such as code generation.

## Acknowledgement

This work is partially supported by the National Key R&D Program of China (NO.2022ZD0160100).

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

## Outline

A. **Supplementary Experiments**

- A.1. Training Set Size in the Augmentation Stage (Table 6)
- A.2. The Selection of Mapping Layers (Table 7)
- A.3. The Usage of Encoder-Decoder Model (Table 8)
- A.4. Quantitative Analysis on Representation Space Changes (Table 9)
- A.5. Translation Performance (Table 10 and Figure 5)
- A.6. Comparison with Enhanced Full Fine-Tuning using Translation Data (Table 11)
- A.7. Inference Speed (Table 12)

B. **Complete Results**

- B.1. Ablation of Mapping Stage (Table 13)
- B.2. Ablation of Augmentation Stage (Table 14)
- B.3. Comparing Replacement and Augmentation (Table 15)
- B.4. Complete Experiments on the MGSM Dataset (Table 16)

C. **Limitations**

D. **Other Tables**

- Table 17. Dataset Statistics
- Table 18. Prompt Templates
- Table 19. Examples of Training Data

## A    Supplementary Experiments

Table 6: Influence of training set size (per language) used in the augmentation stage. Lrl., Hrl., and Avg. represent the average accuracy across low-resource languages, high-resource languages, and all languages, respectively. Referring to Shi et al. [2023], we regard Bn, Th, and Sw as low-resourse languages, and regard the remaining languages as high-resource languages.

| MGSM | Bn | Th | Sw | Ja | Zh | De | Fr | Ru | Es | En | Lrl. | Hrl. | Avg. |
|---|---|---|---|---|---|---|---|---|---|---|---|---|---|
| 0 | 26.8 | 26.8 | 28.8 | 31.2 | 39.6 | 54.0 | 52.4 | 46.0 | 55.2 | 65.2 | 27.5 | 49.1 | 42.6 |
| 100 | 28.4 | 31.6 | 31.6 | 34.0 | 35.6 | 56.8 | 54.0 | 48.0 | 52.8 | 62.4 | 30.5 | 49.1 | 43.2 |
| 1,000 | 36.8 | 43.2 | 37.2 | 40.8 | 41.2 | 57.6 | 52.0 | 52.8 | 52.0 | 62.0 | 39.1 | 51.2 | 47.6 |
| 5,000 | 44.0 | 50.4 | 48.0 | 42.8 | 45.6 | 56.0 | 57.6 | 58.4 | 58.4 | 66.0 | 47.5 | 55.0 | 52.7 |
| 10,000 | 48.4 | 53.6 | 52.8 | **54.8** | **58.4** | 54.8 | 54.8 | 59.2 | 58.8 | 66.4 | 51.6 | 58.2 | 56.0 |
| 20,000 | 45.6 | **54.0** | 54.4 | 52.4 | 53.6 | 60.4 | 54.4 | 58.8 | **59.2** | **68.8** | 51.3 | 58.2 | 55.9 |
| 30,000 | 50.4 | 52.8 | **57.2** | 54.4 | 53.6 | **61.2** | **57.6** | 60.8 | 58.4 | 66.8 | **53.5** | **59.0** | **57.3** |
| 40,000 | **51.6** | 49.6 | 54.4 | 50.0 | 54.0 | 57.2 | 56.0 | **62.8** | 58.8 | 66.4 | 51.9 | 57.9 | 56.1 |

### A.1    Training Set Size in the Augmentation Stage

We randomly sampled different size of training set to train MindMerger-Soft on the mathematical reasoning task and reported the results on MGSM in Table 6. Despite only 100 samples per language is used, MindMerger-Soft can be improved by 3.0% in low-resource languages compared to without augmentation stage. The training data size has a significant impact on the performance of MindMerger within the data size range of 10,000. MindMerger-Soft achieves the best result when using 30,000 samples per language, and better results may require enhancing the model's built-in capabilities.. Considering the query training data for various tasks can be easily obtained from public translation models, and training on a scale of tens of thousands is cost-effective, MindMerger can be readily adapted to a wider range of tasks.

Table 7: The selection of mapping layers. # Parm represents parameters size of the mapping layers. Lrl., Hrl., and Avg. represent the average accuracy across low-resource languages, high-resource languages, and all languages, respectively. Referring to Shi et al. [2023], we regard Bn, Th, and Sw as low-resourse languages, and regard the remaining languages as high-resource languages.

| MGSM | # Parm | Bn | Th | Sw | Ja | Zh | De | Fr | Ru | Es | En | Lrl. | Hrl. | Avg. |
|---|---|---|---|---|---|---|---|---|---|---|---|---|---|---|
| Linear | 4 M | 50.8 | 50.0 | 51.2 | 47.6 | 51.2 | 60.0 | 58.0 | 56.4 | **61.6** | 66.0 | 50.7 | 57.3 | 55.3 |
| 2 layers MLP | 10 M | 50.4 | **52.8** | **57.2** | **54.4** | 53.6 | **61.2** | 57.6 | **60.8** | 58.4 | 66.8 | **53.5** | **59.0** | **57.3** |
| 3 layers MLP | 14 M | **51.6** | 50.0 | 51.2 | 49.6 | **54.4** | 58.0 | **58.4** | 60.0 | 61.2 | **69.2** | 50.9 | 58.7 | 56.4 |
| QFormer | 42 M | 1.6 | 0.8 | 1.6 | 17.2 | 22.4 | 34.0 | 34.4 | 26.0 | 36.8 | 52.8 | 1.3 | 31.9 | 22.8 |

## A.2 The Selection of Mapping Layers

We experimented with the performance of MindMerger-Soft when using different mapping layers in Table 7. The 2 layers MLP used in the main experiment achieved the best results, while the effect of Linear will be lower in comparison. This may be the insufficient space transfer capability of Linear due to the small number of parameters. As for QFormer [Li et al., 2023], which converts variable-length input into fixed-length hidden states output, is a popular model for model merging in computer vision, but it does not work well on MindMerger-Soft. This could be attributed to the multilingual capabilities of LLMs, which may render features diverging too much from the original expression less readily accepted.

Table 8: The usage of encoder-decoder model (M2M100-1.2B). Lrl., Hrl., and Avg. represent the average accuracy across low-resource languages, high-resource languages, and all languages, respectively. Referring to Shi et al. [2023], we regard Bn, Th, and Sw as low-resourse languages, and regard the remaining languages as high-resource languages.

| MGSM | Bn | Th | Sw | Ja | Zh | De | Fr | Ru | Es | En | Lrl. | Hrl. | Avg. |
|---|---|---|---|---|---|---|---|---|---|---|---|---|---|
| Encoder | **50.4** | **52.8** | **57.2** | **54.4** | **53.6** | **61.2** | **57.6** | **60.8** | **58.4** | 66.8 | **53.5** | **59.0** | **57.3** |
| Decoder | 40.4 | 32.0 | 45.6 | 46.8 | 46.0 | 56.8 | 56.0 | 55.2 | 58.0 | **67.6** | 39.3 | 55.2 | 50.4 |
| Encoder + Decoder | 44.0 | 36.4 | 44.8 | 49.6 | 49.6 | 55.6 | 53.6 | 56.0 | **58.4** | 66.8 | 41.7 | 55.7 | 51.5 |

## A.3 The Usage of Encoder-Decoder Model

In all other experiments, for the encoder-decoder model, we only use its encoder part. In this section, we compared the performance of MindMerger-Soft using the decoder. Specifically, we use the translation model M2M100-1.2B autoregressively to generate English text, and take the hidden states corresponding to the generated text, and input them into the mapping layer alone or after concatenating with the output of the encoder. The experimental results are shown in Table 8, where the performance of MindMerger-Soft dropped after adding the hidden states output by the decoder.

Table 9: The changes in the representation space within MindMerger as well as its constituent models the multilingual model (mT5-xl), and the LLM (MetaMath-Llama-7B). Lrl., Hrl., and Avg. represent the average accuracy across low-resource languages, high-resource languages, and all languages, respectively. Referring to Shi et al. [2023], we regard Bn, Th, and Sw as low-resourse languages, and regard the remaining languages as high-resource languages.

| Cosine Similarity | | Bn→En | Th→En | Sw→En | Ja→En | Zh→En | De→En | Fr→En | Ru→En | Es→En | Lrl. | Hrl. | Avg. |
|---|---|---|---|---|---|---|---|---|---|---|---|---|---|
| Multilingual model | Embedding Layer | 0.29 | 0.27 | 0.43 | 0.31 | 0.31 | 0.49 | 0.51 | 0.37 | 0.48 | 0.33 | 0.41 | 0.38 |
| | Last Layer | 0.91 | 0.89 | 0.92 | 0.92 | 0.93 | 0.95 | 0.95 | 0.94 | 0.92 | 0.91 | 0.94 | 0.93 |
| LLM | Embedding Layer | 0.18 | 0.15 | 0.34 | 0.28 | 0.28 | 0.39 | 0.42 | 0.31 | 0.41 | 0.22 | 0.35 | 0.31 |
| | Last Layer | 0.08 | 0.07 | -0.08 | 0.11 | 0.13 | 0.04 | 0.12 | 0.05 | 0.09 | 0.02 | 0.09 | 0.07 |
| MindMerger | Mapping Module | 0.92 | 0.91 | 0.91 | 0.93 | 0.91 | 0.96 | 0.94 | 0.94 | 0.93 | 0.91 | 0.94 | 0.93 |
| | LLM Layer | 0.32 | 0.32 | 0.46 | 0.46 | 0.46 | 0.55 | 0.62 | 0.52 | 0.56 | 0.37 | 0.53 | 0.47 |

| Recall@1 (%) | | Bn→En | Th→En | Sw→En | Ja→En | Zh→En | De→En | Fr→En | Ru→En | Es→En | Lrl. | Hrl. | Avg. |
|---|---|---|---|---|---|---|---|---|---|---|---|---|---|
| Multilingual model | Embedding layer | 1.2 | 3.0 | 16.2 | 1.8 | 4.9 | 28.2 | 34.6 | 10.7 | 29.1 | 6.8 | 18.2 | 14.4 |
| | Last layer | 99.3 | 99.7 | 97.8 | 99.7 | 99.9 | 100 | 100 | 100 | 99.9 | 98.9 | 99.9 | 99.6 |
| LLM | Embedding layer | 0.7 | 1.9 | 5.9 | 3.2 | 4.7 | 30.2 | 41.7 | 8.7 | 27.3 | 2.8 | 19.3 | 13.8 |
| | Last layer | 0.4 | 1.8 | 0.2 | 5.8 | 12.6 | 11.5 | 47.3 | 44.9 | 27.0 | 0.8 | 24.9 | 16.8 |
| MindMerger | Mapping module | 99.4 | 98.9 | 99.4 | 99.6 | 99.9 | 100 | 100 | 99.8 | 99.6 | 99.2 | 99.8 | 99.6 |
| | LLM module | 52.8 | 76.0 | 64.3 | 91.6 | 96.4 | 98.0 | 99.2 | 98.1 | 88.9 | 64.4 | 95.4 | 85.0 |

## A.4 Quantitative Analysis on the Representation Space Changes

We further quantitatively analyzed the changes in the representation space of the models. Two metrics were employ to quantify space relationships: (1) Cosine similarity, indicating the similarity between the same query expressed in different languages, and (2) Recall@1, representing the percentage of top-1 retrieve results from a pool of 1000 English query that correspond to the same query.

The experimental results are shown in Table 9. multilingual model demonstrates the capability to map inputs in different languages to a language-agnostic space. Within this space, the average cosine similarity and Recall@1 from various languages to English reach as high as 0.93 and 99.6. On the contrary, LLM lacks this capability. The average cosine similarity and Recall@1 in the output space of LLM are only 0.07 and 16.8, respectively. The significant disparity in the capabilities to construct language-agnostic representations suggests the potential for LLM to be complemented by multilingual model.

For MindMerger-Soft, consisting of multilingual model and LLM, the mapping layer generates a representation with an average cosine similarity of 0.93 and an average Recall@1 of 99.6. This highly language-agnostic representation as input to LLM effectively helps LLM understand different languages. Compared with the huge difference between the language representations output when using only LLM, the average cosine similarity of LLM output in MindMerger-Soft increased from 0.07 to 0.47, and the average Recall@1 increased from 16.8 to 85.0.

## A.5 Translation Performance

Table 10: Translation results on Flores-101 dataset. spBLEU is used as evaluation metric.

| Flores-101 | Bn→En | Th→En | Sw→En | Ja→En | Zh→En | De→En | Fr→En | Ru→En | Es→En | Avg. |
|---|---|---|---|---|---|---|---|---|---|---|
| Translation Models | | | | | | | | | | |
| MetaMath-Llama-7B (5-shots) | 0.7 | 1.0 | 0.6 | 1.9 | 2.3 | 2.6 | 2.7 | 2.6 | 2.1 | 1.8 |
| M2M100-418M | 25.1 | 18.2 | 26.6 | 20.2 | 21.2 | 36.4 | 39.0 | 28.4 | 25.8 | 26.8 |
| M2M100-1.2B | 27.7 | 23.9 | 34.1 | 24.8 | 26.7 | 42.9 | 43.9 | 34.3 | 30.0 | 32.0 |
| NLLB-200-1.3B | 36.8 | 30.7 | 43.9 | 28.6 | 30.4 | 45.9 | 47.4 | 38.0 | 34.8 | 37.4 |
| NLLB-3.3B | **38.7** | **33.2** | **46.3** | **30.6** | **32.2** | **47.2** | **48.7** | **39.1** | **36.5** | **39.2** |
| MindMerger-Soft (w/o $T$) | | | | | | | | | | |
| mGPT | 2.7 | 2.3 | 3.8 | 2.6 | 4.7 | 6.7 | 10.3 | 4.7 | 6.7 | 5.0 |
| mBERT | 10.4 | 8.2 | 12.4 | 12.5 | 16.3 | 23.9 | 26.1 | 18.7 | 19.0 | 16.4 |
| XLM-RoBERTa-large | 12.6 | 14.2 | 17.5 | 9.2 | 12.6 | 24.6 | 26.7 | 20.2 | 16.9 | 17.2 |
| mT5-large | 18.0 | 18.1 | 25.1 | 13.8 | 16.3 | 30.4 | 34.3 | 25.3 | 24.6 | 22.9 |
| mT5-xl | 25.9 | 23.5 | 34.1 | 19.8 | 22.3 | 37.3 | 39.1 | 30.7 | 29.0 | 29.1 |
| M2M100-418M | 26.3 | 15.0 | 27.3 | 23.8 | 25.6 | 36.8 | 39.4 | 31.3 | 29.6 | 28.3 |
| M2M100-1.2B | 26.6 | 20.0 | 32.6 | 24.5 | 27.0 | 39.3 | 41.7 | 33.3 | 31.2 | 30.7 |
| NLLB-200-1.3B | 30.1 | 26.4 | 38.5 | 21.8 | 24.4 | 38.3 | 41.8 | 32.1 | 30.3 | 31.5 |
| NLLB-200-3.3B | **34.5** | **29.5** | **43.9** | **26.6** | **29.0** | 43.2 | **45.2** | **35.8** | **31.8** | **35.5** |

In Table 10, we used the translation dataset Flores-101 [Fan et al., 2021] as the evaluation set and presented the spBLEU performance of various versions of MindMerger-Soft (using MetaMath-Llama-7B as backbone) trained on mapping stage. In contrast to the poor performance of the same LLM, MetaMath-Llama, MindMerger-Soft greatly boost it translation capabilities. Remarkably, the performance of MindMerger, based on the M2M100-418M encoder part, even surpasses the M2M100-418M translation model itself. Although the encoder-only models mBERT and XLM-RoBERTa-large do not have translation capabilities, MindMerger can still learn translation capabilities based on them.

We also compared the performance of MindMerger with the full fine-tuning method. We conducted experiments across four translation directions using two different sizes of training data, each covering translations between nine pairs of languages, with the comparison of average scores presented in Figure 5. It can be observed that MindMerger has significantly enhanced the translation performance across all settings, including X-En, En-X, X-Zh, and Zh-X, resulting in improvements ranging from 3.7% to 7.5%. Although MindMerger primarily enhances the understanding capabilities of LLMs without directly improving their generation capabilities, its performance on translation task demonstrates that it can also benefit generation tasks. Given that understanding is the basis for generation, MindMerger represents a crucial step towards enhancing the multilingual generation capabilities of LLMs.

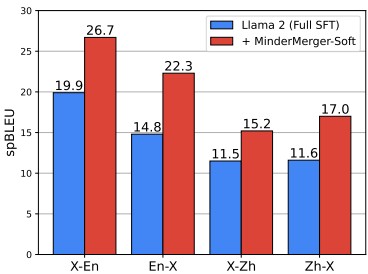
(a) 100K training pairs per languages

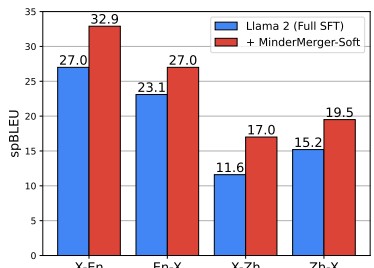
(b) 200K training pairs per languages

Figure 5: Comparison the translation performance of MindMerger and fully supervised fine-tuning method on the Flores-101 dataset. spBLEU is used as evaluation metric.

Table 11: Comparison the effectiveness of extra translation data for full fine-tuning and MindMerger-Soft on the MGSM dataset. Three types of training data are used: (a) translation data, (b) English task data, and (c) query translation task data. → represents the usage order of the training data.

| MGSM | Bn | Th | Sw | Ja | Zh | De | Fr | Ru | Es | En | Lrl. | Hrl. | Avg. |
|---|---|---|---|---|---|---|---|---|---|---|---|---|---|
| (c) | 33.2 | 36.8 | 34.8 | 36.0 | 37.2 | 39.6 | 39.6 | 38.4 | 37.6 | 40.4 | 34.9 | 38.4 | 37.4 |
| (a) → (c) | 36.4 | 36.0 | 37.2 | 31.6 | 34.4 | 42.4 | 40.4 | 40.4 | 42.4 | 41.2 | 36.5 | 39.0 | 38.2 |
| (b) | 6.8 | 7.2 | 6.8 | 36.4 | 38.4 | 55.2 | 54.4 | 52.0 | 57.2 | 68.8 | 6.9 | 51.8 | 38.3 |
| (a) → (b) | 21.2 | 21.2 | 24.4 | 38.8 | 34.4 | 52.8 | 56.8 | 52.4 | 57.6 | 61.6 | 22.3 | 50.6 | 42.1 |
| (b) → (c) | 33.2 | 40.0 | 42.0 | 42.0 | 42.0 | 45.2 | 44.8 | 45.2 | 48.0 | 52.0 | 38.4 | 45.6 | 43.4 |
| (a) → (b) → (c) | 33.6 | 40.4 | 40.4 | 41.2 | 37.6 | 38.4 | 41.2 | 44.4 | 41.6 | 47.6 | 38.1 | 41.7 | 40.6 |
| (c) → (b) | 40.8 | 46.4 | 45.2 | 44.8 | 50.4 | 52.8 | 52.8 | 52.0 | 56.4 | 59.6 | 44.1 | 52.7 | 50.1 |
| (a) → (c) → (b) | 40.8 | 50.0 | 50.4 | 47.6 | 52.8 | 54.8 | 54.0 | 56.0 | 50.0 | 54.8 | 47.1 | 52.9 | 51.1 |
| MindMerger-Soft | **50.4** | **52.8** | **57.2** | **54.4** | **53.6** | **61.2** | **57.6** | **60.8** | **58.4** | **66.8** | **53.5** | **59.0** | **57.3** |

## A.6 The Influence of Translation Data for Full Fine-Tuning

Recall that the training of MindMerger-Soft involves three types of data: (a) translation data, (b) English task data, and (c) query translation task data. Therefore, we compared the effects of full fine-tuning LLMs with the same data to demonstrate that MindMerger-Soft can utilize data more efficiently. The experimental results presented in Table 11 show that, regardless of how the training data are utilized, MindMerger-Soft achieves an average accuracy that is at least 6.2% higher than the full fine-tuned methods. Using translation data as the initial stage of training can enhance performance in 3 out of 4 full fine-tuning settings ((a) → (c), (a) → (b), and (a) → (c) → (b)). The benefits of translation data for MindMerger-Soft are significantly greater than full fine-tuning methods. Specifically, with the assistance of translation data, MindMerger-Soft improved the average accuracy by 5.5% (refer to B.1), while the improvements for full fine-tuning are 0.8%, 3.8%, and 1.0%, respectively.

## A.7 Inference Speed

Table 12: Inference speed and generation length on the MGSM dataset.

| MGSM | Bn | Th | Sw | Ja | Zh | De | Fr | Ru | Es | En | Avg. |
|---|---|---|---|---|---|---|---|---|---|---|---|
| Inference time per sample (s) | | | | | | | | | | | |
| QAlign | **3.8** | 4.2 | **3.6** | 4.1 | 4.1 | 3.8 | 3.6 | **3.6** | 3.8 | **3.5** | 3.8 |
| MindMerger-Soft | 3.9 | **3.6** | **3.6** | **3.5** | **3.7** | **3.5** | **3.5** | 3.9 | **3.7** | 3.6 | **3.7** |
| The average length of generated text (token) | | | | | | | | | | | |
| QAlign | 126.3 | 139.9 | **118.3** | 135.9 | 135.9 | 126.5 | 120.2 | **119.8** | 128.2 | **117.6** | 126.9 |
| MindMerger-Soft | **122.3** | **121.4** | 121.9 | **121.8** | **126.7** | **117.6** | **116.3** | 133.2 | **125.8** | 120.3 | **122.8** |

We compared the inference speed of MindMerger and the single LLM model QAlign on the MGSM dataset with a batch size of 1. As shown in the upper part of Table 12, the average inference speed for MindMerger-Soft is even faster than QAlign, where MindMerger taking an average of 3.7s to infer a sample, compared to QAlign's 3.8s. This discrepancy arises because the text generated by QAlign is slightly longer than that produced by MindMerger-Soft, as indicated in the lower part of Table 12.

# B   Complete Results

Table 13: Ablation of mapping stage.Lrl., Hrl., and Avg. represent the average accuracy across low-resource languages, high-resource languages, and all languages, respectively. Referring to Shi et al. [2023], we regard Bn, Th, and Sw as low-resourse languages, and regard the remaining languages as high-resource languages.

| MGSM | Bn | Th | Sw | Ja | Zh | De | Fr | Ru | Es | En | Lrl. | Hrl. | Avg. |
|---|---|---|---|---|---|---|---|---|---|---|---|---|---|
| MindMerger (M2M100) | **49.6** | **52.4** | **53.2** | **52.8** | **54.4** | **60.0** | 56.4 | **60.0** | 58.0 | 66.0 | **51.7** | 58.2 | **56.3** |
| w/o mapping Stage | 29.2 | 32.4 | 32.8 | 52.0 | 52.8 | 58.4 | **59.6** | 57.2 | **60.0** | **68.4** | 31.5 | **58.3** | 50.3 |
| MindMerger (mT5) | **50.4** | **52.8** | **57.2** | **54.4** | **53.6** | **61.2** | 57.6 | **60.8** | 58.4 | **66.8** | **53.5** | **59.0** | **57.3** |
| w/o mapping Stage | 41.6 | 35.2 | 40.8 | 46.8 | 51.6 | 57.6 | **60.4** | 57.2 | **61.2** | 66.4 | 39.1 | 57.3 | 51.8 |

| MSVAMP | Bn | Th | Sw | Ja | Zh | De | Fr | Ru | Es | En | Lrl. | Hrl. | Avg. |
|---|---|---|---|---|---|---|---|---|---|---|---|---|---|
| MindMerger (M2M100) | **51.7** | **56.0** | **55.2** | **57.2** | **57.7** | **63.1** | **62.1** | **60.2** | **63.4** | 64.3 | **54.3** | **61.1** | **59.1** |
| w/o mapping Stage | 34.1 | 39.9 | 32.4 | 53.7 | 53.6 | 58.6 | 57.5 | 55.6 | 57.9 | **64.6** | 35.5 | 57.4 | 50.8 |
| MindMerger (mT5) | **52.0** | **53.4** | **54.0** | **59.0** | **61.7** | **64.1** | **64.0** | **63.3** | **65.0** | **67.7** | **53.1** | **63.5** | **60.4** |
| w/o mapping Stage | 40.4 | 42.2 | 43.7 | 54.2 | 54.2 | 60.3 | 57.8 | 55.4 | 60.0 | 62.7 | 42.1 | 57.8 | 53.1 |

## B.1   Ablation of Mapping Stage

The complete experimental results are shown in Table 13. The average accuracy of MindMerger-Soft decreased by 6.0% and 5.5% on the MGSM dataset, and by 8.3% and 7.3% on the MSVAMP dataset, demonstrating the effectiveness of this stage. The mapping stage is particularly crucial for enhancing the capabilities of low-resource languages. After ablating this stage, the average accuracy for low-resource languages drops 20.2% and 14.4% on the MGSM dataset, and 18.8% and 11.0% on the MSVAMP dataset. The mapping stage can also be helpful for high-resource languages. After removing the mapping stage, on the MSVAMP dataset, the average accuracy of high-resource languages dropped by 3.7% and 5.7% on the two MindMerger-Soft implementations.

Table 14: Ablation of augmentation stage. Lrl., Hrl., and Avg. represent the average accuracy across low-resource languages, high-resource languages, and all languages, respectively. Referring to Shi et al. [2023], we regard Bn, Th, and Sw as low-resourse languages, and regard the remaining languages as high-resource languages.

| MGSM | Bn | Th | Sw | Ja | Zh | De | Fr | Ru | Es | En | Lrl. | Hrl. | Avg. |
|---|---|---|---|---|---|---|---|---|---|---|---|---|---|
| MonoReason | 7.6 | 5.6 | 5.2 | 34.0 | 45.2 | 54.0 | 56.8 | 51.6 | 58.8 | 65.5 | 6.1 | 52.3 | 38.4 |
| MindMerger (M2M100) | **49.6** | **52.4** | **53.2** | **52.8** | **54.4** | **60.0** | **56.4** | **60.0** | **58.0** | **66.0** | **51.7** | **58.2** | **56.3** |
| w/o augmentation Stage | 25.6 | 23.2 | 26.4 | 37.2 | 40.8 | 51.2 | 51.6 | 49.2 | 56.0 | 64.0 | 25.1 | 50.0 | 42.5 |
| MindMerger (mT5) | **50.4** | **52.8** | **57.2** | **54.4** | **53.6** | **61.2** | **57.6** | **60.8** | **58.4** | **66.8** | **53.5** | **59.0** | **57.3** |
| w/o augmentation Stage | 26.8 | 26.8 | 28.8 | 31.2 | 39.6 | 54.0 | 52.4 | 46.0 | 55.2 | 65.2 | 27.5 | 49.1 | 42.6 |

| MSVAMP | Bn | Th | Sw | Ja | Zh | De | Fr | Ru | Es | En | Lrl. | Hrl. | Avg. |
|---|---|---|---|---|---|---|---|---|---|---|---|---|---|
| MonoReason | 15.0 | 17.1 | 15.4 | 51.9 | 54.4 | 60.9 | 62.2 | 59.3 | 63.3 | 65.5 | 15.8 | 59.6 | 46.2 |
| MindMerger (M2M100) | **51.7** | **56.0** | **55.2** | **57.2** | **57.7** | **63.1** | **62.1** | **60.2** | **63.4** | 64.3 | **54.3** | **61.1** | **59.1** |
| w/o augmentation Stage | 29.4 | 27.6 | 28.2 | 51.2 | 52.5 | 60.4 | 60.6 | 58.2 | 58.8 | **64.3** | 28.4 | 58.0 | 49.1 |
| MindMerger (mT5) | **52.0** | **53.4** | **54.0** | **59.0** | **61.7** | **64.1** | **64.0** | **63.3** | **65.0** | **67.7** | **53.1** | **63.5** | **60.4** |
| w/o augmentation Stage | 37.5 | 35.2 | 37.5 | 52.8 | 53.2 | 61.3 | 61.7 | 57.6 | 63.3 | 64.0 | 36.7 | 59.1 | 52.4 |

## B.2   Ablation of Augmentation Stage

The complete results are shown in Table 14. Even without the mapping stage, MindMerger-Soft solely training in general bilingual pairs can still improve the performance of LLMs in low-resource languages. For the MGSM and MSVAMP datasets, the average accuracy of MindMerger-Soft in low-resource languages increases by MonoReason 19.0% and 12.6% based on M2M100-1.2B, and increases by MonoReason 21.4% and 20.9% based on mT5-xl. With the help of the augmentation stage, the reasoning capabilities of MindMerger-Soft in low-resource languages are further enhanced. Comparing to the MindMerger-Soft without augmentation stage in the low-resource languages, augmentation stage brings improvements of 26.6% and 25.9% based on M2M100-1.2B, and 26.0% and 16.4% based on mT5-xl. Moreover, augmentation stage is also valueable for high-resource languages, bringing improvements of 3.1% based on M2M100-1.2B and 4.4% based on mT5-xl for the MSVAMP dataset in the average accuracy across high-resource languages.

Table 15: Comparison between replacement (w/o $T$) and augmentation. Lrl., Hrl., and Avg. represent the average accuracy across low-resource languages, high-resource languages, and all languages, respectively. Referring to Shi et al. [2023], we regard Bn, Th, and Sw as low-resourse languages, and regard the remaining languages as high-resource languages.

| MGSM | Bn | Th | Sw | Ja | Zh | De | Fr | Ru | Es | En | Url. | Hrl. | Avg. |
|---|---|---|---|---|---|---|---|---|---|---|---|---|---|
| MindMerger-Soft (M2M100) | 49.6 | **52.4** | 53.2 | 52.8 | **54.4** | **60.0** | **56.4** | **60.0** | 58.0 | **66.0** | **51.7** | **58.2** | **56.3** |
| Replacement (w/o $T$) | **50.8** | 44.4 | 50.8 | **54.4** | 50.4 | 57.2 | 55.2 | 58.0 | **61.2** | 63.2 | 48.7 | 57.1 | 54.6 |
| MindMerger-Soft (mT5) | 50.4 | 52.8 | **57.2** | **54.4** | **53.6** | **61.2** | 57.6 | **60.8** | 58.4 | **66.8** | **53.5** | **59.0** | **57.3** |
| Replacement (w/o $T$) | **52.2** | **53.6** | 54.8 | 46.8 | 52.0 | 59.2 | **58.0** | 58.4 | **60.0** | 63.6 | **53.5** | 56.9 | 55.9 |

| MSVAMP | Bn | Th | Sw | Ja | Zh | De | Fr | Ru | Es | En | Url. | Hrl. | Avg. |
|---|---|---|---|---|---|---|---|---|---|---|---|---|---|
| MindMerger-Soft (M2M100) | **51.7** | **56.0** | **55.2** | **57.2** | **57.7** | **63.1** | **62.1** | **60.2** | **63.4** | **64.3** | **54.3** | **61.1** | **59.1** |
| Replacement (w/o $T$) | 50.0 | 48.5 | 52.3 | 55.7 | 57.2 | 60.4 | 59.7 | 55.9 | 57.9 | 58.5 | 50.3 | 57.9 | 55.6 |
| MindMerger-Soft (mT5) | **52.0** | 53.4 | 54.0 | **59.0** | **61.7** | **64.1** | **64.0** | **63.3** | **65.0** | **67.7** | 53.1 | **63.5** | **60.4** |
| Replacement (w/o $T$) | 51.7 | **54.7** | **54.3** | 54.8 | 56.5 | 61.1 | 59.7 | 58.5 | 61.1 | 60.8 | **53.6** | 58.9 | 57.3 |

## B.3 Comparing Replacement and Augmentation

The complete experimental results are shown in Table 15. It can be observed that the augmentation strategy is more significantly helpful in improving high-resource language reasoning capabilities than low-resource languages. Compared with the replacement version of MindMerger-Soft, the augmentation strategy improves the average accuracy by 1.1%, 2.1%, 3.2%, and 4.6% across high-resource languages in the four experimental results. This suggests that, in contrast to the replacement strategy that neglects these capabilities, the augmentation strategy effectively takes advantage of them, thereby unlocking the full potential of LLMs. For low-resource languages, the augmentation strategy also brings improvements of 3.0% and 4.0% for MindMerger-Soft based on M2M100-1.2B, but on par with the replacement strategy for MindMerger-Soft based on mT5-xl. It could be the multilingual capabilities of mT5-xl on low-resource languages are significantly stronger than LLMs that it can completely replace LLMs, while the slightly weaker multilingual capabilities of M2M100-1.2B still need to be complementary to LLMs.

## B.4 Complete Experiments on the MGSM Dataset

Table 16: Results on the MGSM dataset. Lrl., Hrl., and Avg. represent the average accuracy across low-resource languages, high-resource languages, and all languages, respectively. We regard Te, Bn, Th, and Sw as low-resourse languages, and regard the remaining languages as high-resource.

| | Te | Bn | Th | Sw | Ja | Zh | De | Fr | Ru | Es | En | Lrl. | Hrl. | Avg. |
|---|---|---|---|---|---|---|---|---|---|---|---|---|---|---|
| xCoT [Chai et al., 2024] | 42.8 | 40.4 | 49.2 | 48.4 | 50.0 | 50.0 | 47.2 | 47.2 | 47.2 | 48.8 | 48.4 | 45.2 | 49.1 | 47.2 |
| LangBridge [Yoon et al., 2024] | 34.8 | 42.8 | 50.4 | 43.2 | 40.0 | 45.2 | 50.8 | 52.4 | 56.4 | 58.0 | 63.2 | 42.8 | 52.3 | 48.8 |
| MindMerger-Soft | **52.8** | **52.0** | **59.2** | **56.8** | **51.2** | **55.2** | **61.2** | **55.2** | **61.6** | **62.4** | **66.0** | **55.2** | **59.0** | **57.6** |

In the above, we followed the previous work [Chen et al., 2023, Zhu et al., 2024] and experimented with ten languages on the MGSM dataset. However, it is important to note that the MGSM dataset contains a total of eleven languages, including Telugu (Te). Therefore, we presented the evaluation results of MindMerger on the complete MGSM dataset in Table 16. It can be observation that MindMerger-Soft is still the best model on Te, achieving a 10.0% improvement over xCoT [Chai et al., 2024] and a 20.0% improvement over LangBridge on that language, as well as an average 10.4% improvement over xCoT and an 8.8% improvement over LangBridge across all languages.

## C Limitations

Although our experimental results show that merging external and built-in language understanding capabilities can help improve the model's multilingual reasoning capabilities, it is unclear the effective boundaries of this mechanism. On one hand, a multilingual model with significantly weaker language capabilities compared to LLMs might not be able to assist LLMs effectively. On the other hand, LLMs may not be able to provide additional built-in knowledge for a multilingual model whose language capabilities are much stronger. Therefore, more experiments are needed to explore the effective boundaries of MindMerger.

# D    Other Tables

Table 17: Dataset statistics. # Train, # Test, and # Lang refer to the size of query translation training set per language, the size of multilingual test set per language, and the number of language we evaluated for each dataset, respectively.

| Dataset | Task | # Train | # Test | # Lang |
|---------|------|---------|--------|--------|
| MGSM | Math | 30,000 | 250 | 10 |
| MSVAMP | Math | 30,000 | 1,000 | 10 |
| X-CSQA | Commonsense | 8,888 | 1,000 | 16 |
| XNLI | NLI | 2,490 | 5,010 | 15 |

Table 18: The prompts of mathematical reasoning task.

**MonoReason & MultiReason & QAlign**

Below is an instruction that describes a task. Write a response that appropriately completes the request. ### Instruction: {query} ### Response: Let's think step by step.

**Translate-En**

Below is an instruction that describes a task. Write a response that appropriately completes the request. ### Instruction: {translated query} ### Response: Let's think step by step.

**MindMerger-Hard**

Below is an instruction that describes a task. Write a response that appropriately completes the request. ### Instruction: {query} ### Translated instruction: {translated query}### Response: Let's think step by step.

Table 19: Examples of training data.

**Mapping Stage**

**Input (Zh):** 文档中没有元素属于该组时，该名称会被作为根结点显示在结构树上。
**Output:** It will be shown in the structure tree as a top node when there are no elements belonging to this group in the document.

**Augmentation Stage (Math)**

**Input (Zh):** 伯特每天都填报纸上的每日填字游戏。他每两周就用完一支铅笔。平均而言，他用完一支铅笔需要1050个字。每个填字游戏平均有多少个字？
**Input (En):** Bert fills out the daily crossword puzzle in the newspaper every day. He uses a pencil to fill out the puzzles every two weeks. On average, it takes him 1050 words to use up a pencil. How many words are in each crossword puzzle on average?
**Output:** If Bert uses up a pencil to fill out the puzzles every two weeks and it takes him 1050 words to use up a pencil, then he must be filling out 1050 words of crossword puzzles every two weeks. To find out how many words are in each daily crossword puzzle, we need to divide the total number of words (1050) by the number of days in two weeks (14). So, there are 1050/14 = 75 words in each daily crossword puzzle on average. #### The answer is: 75

**Augmentation stage (X-CSQA)**

**Input (Zh):** 酒后驾车的人可能会被指控什么？(A)惩罚 (B)逮捕 (C)车祸 (D)胡言乱语 (E)酒后驾驶
**Input (En):** What is someone operating a vehicle likely to be accused of after becoming inebriated? (A) punish (B) arrest (C) automobile accidents (D) talking nonsense (E) drunk driving
**Output:** E

**Augmentation stage (XNLI)**

**Input (Zh):** Premise: 她不太明白。Hypothesis: 事实上，她没有理解。
**Input (En):** Premise: She doesn't really understand. Hypothesis: Actually, she doesn't get it.
**Output:** Entailment

