# OpenReview forum: "MindMerger: Efficiently Boosting LLM Reasoning in non-English Languages"
_NeurIPS.cc/2024/Conference — NeurIPS 2024 poster_

### Official Review · Reviewer_uQ8B · 2024-06-20

**Soundness:** 4
**Presentation:** 3
**Contribution:** 3
**Rating:** 6
**Confidence:** 4

**Summary:**

The paper introduces MergeMinds, a method that integrates LLMs with multilingual models to enhance reasoning capabilities across multiple languages. Through a two-step training scheme, MergeMinds effectively boosts performance in multilingual reasoning and language understanding tasks, particularly excelling in low-resource languages, as demonstrated by significant accuracy improvements on various datasets.

**Strengths:**

1. This work propose MergeMinds method offers a novel solution by merging LLMs with external multilingual capabilities and wo-step training process, enhancing the robustness of the model.
2. The method shows substantial improvements in multilingual reasoning and language understanding tasks, particularly with notable gains in low-resource languages, demonstrating the effectiveness of the approach.

**Weaknesses:**

1. From my perspective, the motivation of MergeMinds is extremely similar with LangBridge, both of which ultilize the mapping layer to map multilingual encoder to exsisting MLLM.
2. In fact, the work of this paper has already been explored on multi-modal large models a long time ago, such as InstructBlip's Qformer and LLaVA's two-stage alignment training.

**Questions:**

1. It appears that the Te language is not included in the MGSM. Could you explain the reason for this omission?
2. I noticed that some work similar to xcot was referenced but not included in the comparative table. What was the rationale behind this decision?
3. In my view, does this work adapt the paradigm from InstructBlip's Qformer and LLaVA's two-stage alignment training to a multilingual setting?
4. Missing Reference [1-3]

[1] MAPO: Advancing Multilingual Reasoning through Multilingual-Alignment-as-Preference Optimization

[2] The Power of Question Translation Training in Multilingual Reasoning: Broadened Scope and Deepened Insights

[3] Multilingual large language model: A survey of resources, taxonomy and frontiers

**Limitations:**

The authors have adequately addressed the limitations.

---

> ### Author Rebuttal · Authors · 2024-08-07
>
> Thank you for your encouragement of our work. We reply to your questions as follows.
>
> **W1: From my perspective, the motivation of MergeMinds is extremely similar with LangBridge, both of which utilize the mapping layer to map multilingual encoder to existing MLLM.**
>
> - In related works about MLLM, LLM does not have visual capabilities at all. Compared with this, the most significant difference in our research is that LLM has built-in multilingual capabilities, .
> - Based on this difference, we need to explore whether there is a more efficient training solution than MLLM to effectively activate the multilingual capabilities built-in LLM.
> - LangBridge is a solution like the works in MLLM that only considers integrating encoder to obtain capabilities that are not available in LLM. LangBridge does not consider boosting the built-in multilingual capabilities of LLM, which results in its performance being significantly lower than MindMerger. It lags behind MindMerger by 8.0% and 7.1% in low-resource languages and all languages in the MGSM dataset. Therefore, different from the existing work of MLLM, with MindMerger as a starting point, there are still many challenges to explore in the future to stimulate the multilingual as well as other bulit-in capabilities of LLM.
>
>
> **W2, Q3: The work of this paper has already been explored on multi-modal large models. Does this work adapt the paradigm from InstructBlip's Qformer and LLaVA's two-stage alignment training to a multilingual setting?**
>
> - Because the focus of our work is on how to combine multilingual capabilities from external model and LLM itself, we face the challenge of avoiding over-reliance on the capabilities of the external multilingual model during training. Similarly, we also face the challenge of avoiding over-reliance on the multilingual capabilities built into LLM during training.
> - Moreover, our goal is to activate the multilingual capabilities of the LLM rather than to increase new capabilities it does not have, it is possible to explore a more efficient training solution than the MLLM. It can be observed from Table 6 in the paper that MindMerger has surpassed all baselines under the using of only 5,000 training samples per language at the second stage.
> - In addition, the research on integrating multilingual models into LLM is still under-explored and different from the existing work on MLLM. For example, what data to use for training, what structure to use for integration, etc. still need more insightful discussions.
>
> | MGSM       | Te    | Bn    | Th    | Sw    | Ja    | Zh    | De    | Fr    | Ru    | Es    | En    | Lrl   | Hrl   |
> |------------|-------|-------|-------|-------|-------|-------|-------|-------|-------|-------|-------|-------|-------|
> | xCoT       | 42.8  | 40.4  | 49.2  | 48.4  | 50.0  | 50.0  | 47.2  | 49.6  | 50.0  | 48.8  | 48.4  | 45.2  | 49.1  |
> | LangBridge | 34.8  | 42.8  | 50.4  | 43.2  | 40.0  | 45.2  | 50.8  | 52.4  | 56.4  | 58.0  | 63.2  | 42.8  | 52.3  |
> | MindMerger | **52.0**  | **54.0**  | **54.0**  | **55.6**  | **51.6**  | **54.8**  | **60.0**  | **59.6**  | **57.6**  | **63.2**  | **65.2**  | **53.9**  | **58.9**  |
>
>
> **Q1: It appears that the Te language is not included in the MGSM. Could you explain the reason for this omission?**
>
> - This is not a technical issue, since the QAlign and MathOctopus work we followed did not compare Te, so we ignored Te as well.
> - We added the Te experiment in the table above. MindMerger is still the best model on Te, with a 9.2% improvement over xCoT.
>
> **Q2: I noticed that some work similar to xCoT was referenced but not included in the comparative table. What was the rationale behind this decision?**
>
> - We selected the two strongest methods for each of the two categories of baselines: relearning-based and replacement-based.
> - xCoT is a relearning-based method, but its performance is lower than MultiReason (MultiReason-Lora) and QAlign. Considering the page limit of the paper, we did not include it in the comparison.
> - We show the performance of xCoT in the table above, and we consider adding it to the appendix.
>
>
> **Q4: Missing Reference.**
> - Thank you for the reviewers’ kind reminder, we will cite them in the next version of paper.
>
> Best wishes.

---

> ### Comment · Reviewer_uQ8B · 2024-08-12
>
> Thanks for your great response. I have no additional problems.

---

### Official Review · Reviewer_VBLh · 2024-07-09

**Soundness:** 3
**Presentation:** 3
**Contribution:** 2
**Rating:** 6
**Confidence:** 3

**Summary:**

This paper proposes a framework which merges LLMs with the external language understanding capabilities from multilingual models to improve multilingual reasoning performance. Specifically, the authors introduce a two-step training scheme:  they (i) train the framework to embed the multilingual model into the LLM by using translation data, and (ii) further train the framework to collaboratively utilize the built-in capabilities of LLM and the embedded external language capabilities.

**Strengths:**

- This paper proposes a new method to boost the multilingual reasoning of LLMs, which embeds external language understanding capabilities from multilingual models into LLMs.
- Extensive experimental results and analyses on several datasets demonstrate the effectiveness of the proposed method.

**Weaknesses:**

- In Section 3.2, the authors mention that they use translation data and query translation task data generated from public translation models for the two-stage training; while in lines 182-192, it shows that they also use several English task datasets, such as MetaMathQA. It is unclear how they use these data.
- Although the author mentioned that only a small number of parameters are trained during two-stage training,  the paper lacks a comparison with baselines in this apsect. Particularly, the proposed method requires the external multilingual model, which is expected to affect the inference efficiency.
- I know asking for new results might be unrealistic at this stage, but I am curious what would happen if you only used two-stage training for LLM without including an external multilingual model.

**Questions:**

See above.

**Limitations:**

See above.

---

> ### Author Rebuttal · Authors · 2024-08-07
>
> Thank you for the valuable reviews. We have added some experiments and will include them in the next version of paper.
>
> **W1. In Section 3.2, the authors mention that they use translation data and query translation task data generated from public translation models for the two-stage training; while in lines 182-192, it shows that they also use several English task datasets, such as MetaMathQA. It is unclear how they use these data.**
>
> - For all models, we first use English task datasets to fully fine-tune LLM in order to obtain the reason capabilities, and then use other data to train.
> - Taking the mathematical reasoning task as an example, both our model and all baselines are based on MetaMath-Llama (denote as MonoReason in our paper) as the base model, and MetaMath-Llama is fully fine-tuning Llama on the MetaMathQA dataset.
>
> **W2. Although the author mentioned that only a small number of parameters are trained during two-stage training, the paper lacks a comparison with baselines in this apsect. Particularly, the proposed method requires the external multilingual model, which is expected to affect the inference efficiency.**
>
> - **Compared with the parallel encoding input query, the serial generation time is much longer.**
> - As report in the follow table, compared with the single LLM model QAlign under batch size 1, the average inference speed for **MindMerger even faster than QAlign**, where MindMerger spends an average of 3.65s inferring a sample while QAlign uses 3.81.
>
> | Inference time per sample (s) | Bn    | Th    | Sw    | Ja    | Zh    | De    | Fr    | Ru    | Es    | En    | Avg.   |
> |-------------------------------|-------|-------|-------|-------|-------|-------|-------|-------|-------|-------|--------|
> | QAlign                        | 3.77  | 4.21  | 3.57  | 4.09  | 4.13  | 3.84  | 3.61  | 3.62  | 3.79  | 3.48  | 3.81   |
> | MindMerger                    | 3.88  | 3.57  | 3.64  | 3.53  | 3.71  | 3.49  | 3.47  | 3.94  | 3.74  | 3.55  | 3.65   |
>
> - This phenomenon is caused by the length of the text generated by QAlign is slightly larger than that of MindMerger as reported as belows:
>
> | The average length of generated text (token) | Bn      | Th      | Sw      | Ja      | Zh      | De      | Fr      | Ru      | Es      | En      | Avg.     |
> |----------------------------------------------|---------|---------|---------|---------|---------|---------|---------|---------|---------|---------|----------|
> | QAlign                                       | 126.30  | 139.94  | 118.34  | 135.86  | 135.94  | 126.52  | 120.20  | 119.76  | 128.24  | 117.60  | 126.87   |
> | MindMerger                                   | 122.32  | 121.42  | 121.94  | 121.78  | 126.74  | 117.56  | 116.30  | 133.22  | 125.84  | 120.34  | 122.75   |
>
> - As for the cost of the external multilingual model, it is very insignificant. As reported in the below table, due to its small number of parameters, the time used to encode the query of external model is only half of QAlign (0.02s vs 0.04s).
>
> | Encoding time per sample (s)         | Bn    | Th    | Sw    | Ja    | Zh    | De    | Fr    | Ru    | Es    | En    | Avg.   |
> |--------------------------------------|-------|-------|-------|-------|-------|-------|-------|-------|-------|-------|--------|
> | QAlign                               | 0.05  | 0.04  | 0.04  | 0.03  | 0.03  | 0.03  | 0.04  | 0.03  | 0.03  | 0.03  | 0.04   |
> | Multilingual Encoder + Mapping Layer | 0.02  | 0.02  | 0.02  | 0.02  | 0.02  | 0.02  | 0.02  | 0.02  | 0.02  | 0.02  | 0.02   |

---

> ### Author Response · Authors · 2024-08-07
> **(2/2) Response to Reviewer VBLh**
>
> **W3. I know asking for new results might be unrealistic at this stage, but I am curious what would happen if you only used two-stage training for LLM without including an external multilingual model.**
>
> - There are three types of training data in the paper: **(a)** translation data, **(b)** English task data, and **(c)** query translation task data.
> - We experimented with four dataset usage settings (b, c, b->c, c->b) to fully fine-tune Llama and compare the influence of using translation data for training in the first stage (a->b, a->c, a->b->c, a->c->b).
> - The experimental results are shown in the table below. Regardless of the way the training data is used, MindMerger has an average accuracy that is at least 6.2 higher than the fully fine-tuned method.
> - Using translation data as the first stage training data can help improve performance on 3 out of 4 fully fine-tuning settings (a->c, a->b, and a->c->b).
> - The gain of translation data for MindMerger is significantly greater than that of fully fine-tuning Llama. With the help of translation data, MindMerger improved the average accuracy rate by 5.5%, while the improvements for fully fine-tuning Llama were 0.8%, 3.8% and 1.0% respectively.
>
>
> |  | Bn    | Th    | Sw    | Ja    | Zh    | De    | Fr    | Ru    | Es    | En    | Lrl.  | Hrl.  | Avg.   |
> |--------------------------------------------------------------------------------------|-------|-------|-------|-------|-------|-------|-------|-------|-------|-------|-------|-------|--------|
> | c                                                                                    | 33.2  | 36.8  | 34.8  | 36.0  | 37.2  | 39.6  | 39.6  | 38.4  | 37.6  | 40.4  | 34.9  | 38.4  | 37.4   |
> | a->c                                                                                 | 36.4  | 36.0  | 37.2  | 31.6  | 34.4  | 42.4  | 40.4  | 40.4  | 42.4  | 41.2  | 36.5  | 39.0  | 38.2   |
> | b (MonoReason)                                                                       | 6.8   | 7.2   | 6.8   | 36.4  | 38.4  | 55.2  | 54.4  | 52.0  | 57.2  | 68.8  | 6.9   | 51.8  | 38.3   |
> | a->b                                                                                 | 21.2  | 21.2  | 24.4  | 38.8  | 34.4  | 52.8  | 56.8  | 52.4  | 57.6  | 61.6  | 22.3  | 50.6  | 42.1   |
> | b->c (MultiRason)                                                                    | 33.2  | 40.0  | 42.0  | 42.0  | 42.0  | 45.2  | 44.8  | 45.2  | 48.0  | 52.0  | 38.4  | 45.6  | 43.4   |
> | a->b->c                                                                              | 33.6  | 40.4  | 40.4  | 41.2  | 37.6  | 38.4  | 41.2  | 44.4  | 41.6  | 47.6  | 38.1  | 41.7  | 40.6   |
> | c->b                                                                                 | 40.8  | 46.4  | 45.2  | 44.8  | 50.4  | 52.8  | 52.8  | 52.0  | 56.4  | 59.6  | 44.1  | 52.7  | 50.1   |
> | a->c->b                                                                              | 40.8  | 50.0  | 50.4  | 47.6  | 52.8  | 54.8  | 54.0  | 56.0  | 50.0  | 54.8  | 47.1  | 52.9  | 51.1   |
>
>
> Thanks again to the reviewer for the experimental suggestions. We believe that these additional experiments will help further enhance the impact of our work.
>
> Best wishes.

---

> ### Comment · Reviewer_VBLh · 2024-08-12
> **Reviewer Response to Authors' Rebuttal**
>
> Dear authors,
>
> Thanks for your response and update with the results! I decided to raise the soundness score.

---

### Official Review · Reviewer_q66P · 2024-07-12

**Soundness:** 3
**Presentation:** 2
**Contribution:** 3
**Rating:** 6
**Confidence:** 5

**Summary:**

The paper propose a way to improve multilingual reasoning in LLM in own native languages without relying on pivoting methods such as translating to English. The method assume a hypothesis that LLMs have built in knowledge and reasoning abilities in a lower-resource language and not just common language like English. The method plugs in LLMs with the external multilingual models as embedding layers and introduce a training scheme build those new knowledge it. Experiments on MGSM show better accuracy across languages.

**Strengths:**

* The method is pretty novel in ways of incorporating multilingual encoders as additional embedding layers to augment existing embedding layers of the LLMs.
* The experiments are done with many languages comprehensively, including many low-resource languages, and ablation studies, which are good.

**Weaknesses:**

* The method is incomplete and ambiguous on the generation side. It makes sense that injecting multilingual embeddings at the bottom for the input may help the LLM understand the context better, but that does not necessarily make the LLM **generate** in those native language better, especially if the LLMs are weak in producing content in native languages.
   * It is no surprise that the tasks presented in the paper are all "understanding" tasks where the generation workload is light and trivial. Other "generation" tasks, such as summarization, writing, translation... may not fair well with this method. I would highlight if the authors could show good results for the method for these tasks.

* Missing baseline comparison: https://arxiv.org/abs/2306.11372 , https://aclanthology.org/2023.findings-emnlp.826.pdf

* Confusing word choices, such as "mapping stage", "merging" and "augmentation stage". These words often lead to different understanding rather than what are described in the paper.

* "Query translation task" and "query translation data" and many parts of the methodology are not explained clearly. It would be better if there are explicit visual examples or diagrams.

**Questions:**

NA

**Limitations:**

The paper discussed limitations

---

> ### Author Rebuttal · Authors · 2024-08-07
>
> Thank you for your reviews.
>
> The main concern is the performance of our model on the generation task. We believe that adding experiments on the generation task will help further expand the scope of our work.
>
> We are grateful for the feedbacks on some writing improvement and we will revise them in the next version.
>
> The specific responses are as follows:
>
> **W1. The method is incomplete and ambiguous on the generation side. It makes sense that injecting multilingual embeddings at the bottom for the input may help the LLM understand the context better, but that does not necessarily make the LLM generate in those native language better, especially if the LLMs are weak in producing content in native languages.**
>
>
> - Our model can be a key step towards improving LLM multilingual generation capabilities, since **language understanding is the basis of language generation**.
>
> - To response this concern, we added an experimental comparison on the translation task based on Flores-101 dataset. We implemented four settings: X-En, En-X, X-Zh, and Zh-X. In each setting, we compare fully fine-tuned Llama2 (Llama-SFT) with MindMerger using 100 K and 200 K training samples per language, respectively.
>
> - The experimental results are shown in the following four tables. **MindMerger consistently outperforms Llama-SFT in translation quality** under all comparison settings.
>
> - Considering that the top of MindMerger is frozen and not trained, the improvement in generation quality comes from the enhancement of query understanding, which may be a valuable insight for future research on multilingual generation tasks.
>
> | X-En                 | Bn    | Th    | Sw    | Ja    | Zh    | De    | Fr    | Ru    | Es    | Lrl.  | Hrl.  | Avg.   |
> |----------------------|-------|-------|-------|-------|-------|-------|-------|-------|-------|-------|-------|--------|
> | Llama-SFT  (100 K)   | 14.8  | 9.6   | 22.7  | 16.4  | 14.6  | 28.7  | 26.3  | 25.0  | 20.7  | 15.7  | 22.0  | 19.9   |
> | MinderMerger (100 K) | 18.6  | 16.7  | 32.9  | 23.3  | 22.2  | 37.4  | 32.6  | 31.7  | 25.0  | 22.7  | 28.7  | 26.7   |
> | Llama-SFT (200 K)    | 16.0  | 18.8  | 27.9  | 20.2  | 23.6  | 38.1  | 39.3  | 30.4  | 28.7  | 20.9  | 30.1  | 27.0   |
> | MinderMerger (200 K) | 27.1  | 24.5  | 40.2  | 26.0  | 28.9  | 42.5  | 42.4  | 34.1  | 30.1  | 30.6  | 34.0  | 32.9   |
>
>
> | En-X                 | Bn    | Th    | Sw    | Ja    | Zh    | De    | Fr    | Ru    | Es    | Lrl.  | Hrl.  | Avg.   |
> |----------------------|-------|-------|-------|-------|-------|-------|-------|-------|-------|-------|-------|--------|
> | Llama-SFT            | 17.1  | 8.2   | 20.8  | 13.1  | 6.2   | 18.9  | 19.7  | 12.6  | 16.7  | 15.4  | 14.5  | 14.8   |
> | MinderMerger         | 20.2  | 11.9  | 30.1  | 19.2  | 12.4  | 27.6  | 32.2  | 23.2  | 23.7  | 20.7  | 23.1  | 22.3   |
> | Llama-SFT (200 K)    | 23.9  | 15.6  | 26.6  | 18.9  | 16.7  | 25.1  | 35.6  | 21.2  | 24.3  | 22.0  | 23.6  | 23.1   |
> | MinderMerger (200 K) | 24.4  | 17.4  | 32.6  | 23.0  | 20.7  | 31.6  | 40.8  | 26.6  | 26.1  | 24.8  | 28.1  | 27.0   |
>
>
> | X-Zh                 | Bn    | Th    | Sw    | Ja    | De    | Fr    | Ru    | Es    | En    | Lrl.  | Hrl.  | Avg.   |
> |----------------------|-------|-------|-------|-------|-------|-------|-------|-------|-------|-------|-------|--------|
> | Llama-SFT (100 K)    | 7.0   | 5.7   | 6.9   | 12.9  | 15.7  | 13.5  | 14.3  | 11.6  | 16.3  | 6.5   | 14.1  | 11.5   |
> | MinderMerger (100 K) | 10.1  | 8.9   | 10.3  | 16.0  | 18.7  | 18.0  | 17.4  | 15.8  | 21.2  | 9.8   | 17.9  | 15.2   |
> | Llama-SFT (200 K)    | 8.5   | 7.5   | 8.2   | 13.2  | 14.7  | 14.4  | 13.6  | 11.1  | 13.1  | 8.1   | 13.4  | 11.6   |
> | MinderMerger (200 K) | 11.8  | 11.7  | 12.0  | 17.3  | 20.5  | 20.0  | 19.0  | 17.0  | 23.9  | 11.8  | 19.6  | 17.0   |
>
>
> | Zh-X                 | Bn   | Th   | Sw   | Ja   | De   | Fr    | Ru    | Es    | En    | Lrl.  | Hrl.  | Avg.   |
> |----------------------|------|------|------|------|------|-------|-------|-------|-------|-------|-------|--------|
> | Llama-SFT (100 K)    | 9.9  | 3.6  | 7.3  | 12.9 | 11.5 | 14.7  | 9.6   | 11.0  | 12.7  | 6.9   | 12.1  | 10.4   |
> | MinderMerger (100 K) | 14.1 | 7.8  | 13.9 | 19.4 | 18.8 | 24.8  | 17.5  | 17.7  | 25.9  | 11.9  | 20.7  | 17.8   |
> | Llama-SFT (200 K)    | 14.2 | 7.8  | 11.3 | 16.5 | 17.7 | 20.5  | 14.1  | 15.7  | 18.6  | 11.1  | 17.2  | 15.2   |
> | MinderMerger (200 K) | 16.2 | 11.4 | 16.2 | 20.7 | 20.2 | 26.6  | 18.7  | 18.3  | 27.2  | 14.6  | 22.0  | 19.5   |

---

> ### Author Response · Authors · 2024-08-07
> **(2/3) Response to Reviewer q66P**
>
> **W2. Missing baseline comparison XLT.**
>
> - **We have discussed XLT in related work, lines 80-81**, which is a prompt-based approach that brings limited improvements over open source models such as Llama through carefully crafted prompts.
> - Our model and the baselines we compared are all supervised fine-tuning methods. Although the work of XLT is inspiring, prompt-based methods are not in the same category works as our model.
> - We added a comparison between XLT and our model. The experimental results are shown in the following two tables. Under the same base model, MindMerger has a huge lead over XLT.
> - Due to page limitations, we consider adding the experimental results of XLT to the appendix.
>
> |                     | Bn    | Th    | Sw    | Ja    | Zh    | De    | Fr    | Ru    | Es    | En    | Lrl   | Hrl   | Avg.   |
> |---------------------|-------|-------|-------|-------|-------|-------|-------|-------|-------|-------|-------|-------|--------|
> | XLT (Llama2-chat)   | 3.2   | 5.6   | 4.0   | 9.2   | 12.8  | 16.0  | 17.6  | 14.0  | 19.6  | 19.6  | 4.3   | 15.5  | 12.2   |
> | MindMerger (Llama2) | 50.4  | 52.8  | 57.2  | 54.4  | 53.6  | 61.2  | 57.6  | 60.8  | 58.4  | 66.8  | 53.5  | 59.0  | 57.3   |
>
>
> |                       | Bn    | Th    | Sw    | Ja    | Zh    | De    | Fr    | Ru    | Es    | En    | Lrl   | Hrl   | Avg.   |
> |-----------------------|-------|-------|-------|-------|-------|-------|-------|-------|-------|-------|-------|-------|--------|
> | XLT (Llama3-Instruct) | 27.6  | 53.6  | 32.4  | 52.0  | 58.0  | 53.6  | 39.6  | 50.8  | 53.2  | 52.8  | 37.9  | 51.4  | 47.4   |
> | MindMerger (Llama3)   | 64.4  | 65.6  | 66.4  | 62.0  | 68.0  | 71.6  | 72.4  | 73.2  | 72.8  | 75.2  | 65.5  | 70.7  | 69.2   |

---

> ### Author Response · Authors · 2024-08-07
> **(3/3) Response to Reviewer q66P**
>
> **W3. Confusing word choices, such as "mapping stage", "merging" and "augmentation stage". These words often lead to different understanding rather than what are described in the paper.**
>
> - Thank you for your feedback, we will fine-tune the wording and description in the next version based on your comments.
>
>
> **W4. "Query translation task" and "query translation data" and many parts of the methodology are not explained clearly. It would be better if there are explicit visual examples or diagrams.**
>
> - Thank you for your feedback again. We followed the settings of QAlign to translate the query from English to non-English automatically as training data. Below are three examples of query translation task data:
>
> |             | Augmentation stage (Query translation task data)                                                                                                                                                                                                                                                                                                                                                                                                                                                                                                                                         |
> |-------------|------------------------------------------------------------------------------------------------------------------------------------------------------------------------------------------------------------------------------------------------------------------------------------------------------------------------------------------------------------------------------------------------------------------------------------------------------------------------------------------------------------------------------------------------------------------------------------------|
> | Math        | **Input (Zh):** 伯特每天都填报纸上的每日填字游戏。他每两周就用完一支铅笔。平均而言，他用完一支铅笔需要1050个字。每个填字游戏平均有多少个字？ <br> **Input (En):** Bert fills out the daily crossword puzzle in the newspaper every day. He uses a pencil to fill out the puzzles every two weeks. On average, it takes him 1050 words to use up a pencil. How many words are in each crossword puzzle on average? <br> **Training Target:** If Bert uses up a pencil to fill out the puzzles every two weeks and it takes him 1050 words to use up a pencil ... 1050/14 = 75 words in each daily crossword puzzle on average. #### The answer is: 75 |
> | Commonsense | **Input (Zh):** 酒后驾车的人可能会被指控什么? (A)惩罚 (B)逮捕 (C)车祸 (D)胡言乱语 (E)酒后驾驶 <br> **Input (En):** What is someone operating a vehicle likely to be accused of after becoming inebriated? (A) punish (B) arrest (C) automobile accidents (D) talking nonsense (E) drunk driving <br> **Training Target:** E                                                                                                                                                                                                                                                                                          |
> | NLI         | **Input (Zh):** Premise: 她不太明白。Hypothesis: 事实上，她没有理解 <br> Input (En): Premise: She doesn’t really understand. Hypothesis: Actually, she doesn’t get it. <br> **Output:** Entailment                                                                                                                                                                                                                                                                                                                                                                                                      |
>
>
> If you have any questions about our work, we look forward to discussing with you and, if possible, hope that you can improve our score appropriately.
>
> Best wishes.

---

> ### Comment · Area_Chair_868M · 2024-08-12
>
> Dear reviewer -- could you see whether the author response addressed your concern, especially about your comment on how their method can improve *generation* of LLMs? they provided new experiments, as well as a comparison with baseline you proposed. If you can modify the scores based on the response, that'd be very helpful!

---

> > ### Comment · Reviewer_q66P · 2024-08-13
> > **Thanks for the rebuttal**
> >
> > Thanks for the author response, I pump up the scores for the effort!

---

### Official Review · Reviewer_iF4J · 2024-07-24

**Soundness:** 2
**Presentation:** 3
**Contribution:** 2
**Rating:** 5
**Confidence:** 4

**Summary:**

The paper suggests incorporating an embedding block using external multilingual models to improve the models' understanding. Additionally, comprehensive experiments are conducted to demonstrate its efficacy.

**Strengths:**

1. The paper proposes a straightforward and easily implementable method to enhance models' multilingual capability.
2. Experimental results validate its effectiveness.

**Weaknesses:**

The paper lacks a clear understanding of why the method can work.
1. To my understanding, the Embedding Layer of the model merely converts the query from text space to the token/embedding space. This transformation may not be interpreted as "understanding the input" since the information hasn't been operated and combined by the attention layer. However, $X$ after the multilingual model has been "understood" by the model. Therefore, it does not make much sense to concatenate these two embeddings.
2. As $\tilde{X}$ contains the information of query q, $T$ includes duplicated information. This may be why the method can surpass baselines such as QAlign and Translate-En. It would be more convincing if baselines also input duplicated information.
3. This is further proved by Figure (4b), which illustrates that the mapping layer tends to obtain a unified representation for different languages and this is the same as QAlign. Therefore, it is confusing why this method can outperform QAlign.
4. The paper uses Llama2 as the base model. However, its multilingual performance is far from sota. Until submission, I believe Mistral, Gemma, and Llama3 have been released, whose multilingual performance is much better.

**Questions:**

The details of the two training stages lack clarity, specifically regarding the utilization of translation and parallel data, as well as the precise training tasks involved.

**Limitations:**

The authors suggest a limitation on the integration of external multilingual models with the model's inherent understanding capabilities, a concern partly explored through the ablation study.

---

> ### Author Rebuttal · Authors · 2024-08-07
>
> We thank the reviewer for taking the time to provide reviews. However, we would like to clarify some misunderstandings as follows:
>
> **W1. To my understanding, the Embedding Layer of the model merely converts the query from text space to the token/embedding space. This transformation may not be interpreted as "understanding the input" since the information hasn't been operated and combined by the attention layer. However,
> after the multilingual model has been "understood" by the model. Therefore, it does not make much sense to concatenate these two embeddings.**
>
> - Our model is based on the finding that providing LLM with inputs from two representation spaces can improve LLM's multilingual reasoning capability, where one representation comes from the native space of LLM's embedding layer, and the other representation is the mapping of multilingual queries to unified space provided by the multilingual model.
>
> - The two different types of representations are the key to boost better understanding of the query in subsequent layers of LLM.
>
>
> **W2. As contains the information of query q, includes duplicated information. This may be why the method can surpass baselines such as QAlign and Translate-En. It would be more convincing if baselines also input duplicated information.**
>
> - We duplicated the input for all baselines during training and inference. As shown in the below table, **the performance of 4 out of 6 baselines decreased with duplicated input**.
> - Note that our model outperforms all baselines by at least 6.7% and 8.0% across all languages
> and low-resource languages, which far exceeds the improvement of the baseline by duplicated input. Therefore, it can be considered that duplicate input is not the mechanism for improving the performance of our model.
>
> | MetaMath-Llama-7B, MGSM             | Bn    | Th    | Sw    | Ja    | Zh    | De    | Fr    | Ru    | Es    | En    | Lrl   | Hrl   | Avg.   |
> |-------------------------------------|-------|-------|-------|-------|-------|-------|-------|-------|-------|-------|-------|-------|--------|
> | MonoReason                          | 6.8   | 7.2   | 6.8   | 36.4  | 38.4  | 55.2  | 54.4  | 52.0  | 57.2  | 68.8  | 6.9   | 51.8  | 38.3   |
> | MonoReason (duplicated input)       | 8.4   | 7.2   | 4.8   | 33.2  | 42.8  | 53.6  | 53.6  | 53.6  | 52.0  | 63.2  | 6.8   | 50.3  | 37.2   |
> | MultiReason-Lora                    | 29.6  | 35.2  | 28.0  | 52.0  | 54.8  | 59.6  | 58.4  | 62.4  | 59.6  | 64.8  | 30.9  | 58.8  | 50.4   |
> | MultiReason-Lora (duplicated input) | 28.8  | 43.2  | 33.6  | 50.4  | 54.8  | 57.6  | 57.6  | 61.2  | 62.0  | 62.8  | 35.2  | 58.1  | 51.2   |
> | MultiReason-SFT                     | 33.2  | 40.0  | 42.0  | 42.0  | 42.0  | 45.2  | 44.8  | 45.2  | 48.0  | 52.0  | 38.4  | 45.6  | 43.4   |
> | MultiReason-SFT  (duplicated input) | 31.2  | 41.2  | 39.2  | 42.0  | 47.6  | 49.6  | 52.0  | 49.6  | 48.8  | 58.8  | 46.0  | 49.8  | 46.0   |
> | QAlign                              | 32.4  | 39.6  | 40.4  | 44.0  | 48.4  | 54.8  | 56.8  | 52.4  | 59.6  | **68.0**  | 37.5  | 54.9  | 49.6   |
> | QAlign (duplicated input)           | 30.8  | 38.0  | 40.4  | 44.4  | 46.4  | 55.6  | 55.2  | 56.8  | 57.6  | 65.2  | 49.0  | 54.5  | 49.0   |
> | LangBridge                          | 42.8  | 50.4  | 43.2  | 40.0  | 45.2  | 56.4  | 50.8  | 52.4  | 58.0  | 63.2  | 45.5  | 52.3  | 50.2   |
> | LangBridge (duplicated input)       | 34.0  | 42.8  | 42.0  | 33.2  | 35.6  | 53.6  | 52.0  | 51.2  | 57.2  | 61.2  | 39.6  | 49.1  | 46.3   |
> | Translate-En                        | 48.4  | 37.6  | 37.6  | 49.2  | 46.8  | 60.4  | 56.4  | 47.6  | 59.6  | 65.5  | 41.2  | 55.1  | 50.9   |
> | Translate-En (duplicated input)     | 47.6  | 33.6  | 38.4  | 48.4  | 48.8  | 56.0  | 49.2  | 44.8  | 54.8  | 63.2  | 39.9  | 52.2  | 48.5   |
> | **MindMerger**                          | **52.0**  | **53.4**  | **54.0**  | **59.0**  | **61.7**  | **64.1**  | **64.0**  | **63.3**  | **65.0**  | 67.7  | **53.1**  | **63.5**  | **60.4**   |

---

> ### Author Response · Authors · 2024-08-07
> **(2/3) Response to Reviewer iF4J**
>
> **W3. This is further proved by Figure (4b), which illustrates that the mapping layer tends to obtain a unified representation for different languages and this is the same as QAlign. Therefore, it is confusing why this method can outperform QAlign.**
>
> - This reflects that **our model has better capability to obtain unified representation than QAlign**.
> - In Table 9 of the paper, we calculated the cosine similarity and top-1 retrieval recall between representations of the same query expressed in different languages to quantify the model’s capability to obtain a unified representation. The results show that our model significantly improves LLM’s capability to obtain a unified representation.
> - We further compare the last layer pooling representation of QAlign and our method in the following table. It can be observed that although QAlign slightly outperforms MonoReason, the score of cosine similarity and recall@1 are still far behind our model, which proves that our model is better able to obtain unified representations.
>
>  | Cosine Similarity | Bn->En | Th->En | Sw->En | Ja->En | Zh->En | De->En | Fr->En | Ru->En | Es->En | Lrl   | Hrl   | Avg.   |
> |-------------------|--------|--------|--------|--------|--------|--------|--------|--------|--------|-------|-------|--------|
> | MonoReason        | 0.08   | 0.07   | -0.08  | 0.11   | 0.13   | 0.04   | 0.12   | 0.05   | 0.09   | 0.02  | 0.09  | 0.07   |
> | QAlign            | 0.07   | 0.04   | -0.06  | 0.11   | 0.14   | 0.10   | 0.18   | 0.06   | 0.15   | 0.02  | 0.12  | 0.09   |
> | MindMerger        | **0.32**   | **0.32**   | **0.46**   | **0.46**   | **0.46**   | **0.55**   | **0.62**   | **0.52**   | **0.56**   | **0.37**  | **0.53**  | **0.48**   |
>
>
> | Recall@1 (%) | Bn->En | Th->En | Sw->En | Ja->En | Zh->En | De->En | Fr->En | Ru->En | Es->En | Lrl   | Hrl   | Avg.   |
> |--------------|--------|--------|--------|--------|--------|--------|--------|--------|--------|-------|-------|--------|
> | MonoReason   | 0.4    | 1.8    | 0.2    | 5.8    | 12.6   | 11.5   | 47.3   | 44.9   | 27.0   | 0.8   | 24.9  | 16.8   |
> | QAlign       | 2.8    | 4.9    | 0.5    | 12.3   | 26.1   | 21.2   | 61.7   | 37.3   | 41.0   | 2.7   | 33.3  | 23.1   |
> | MindMerger   | **52.8**   | **76.0**   | **64.3**   | **91.6**   | **96.4**   | **98.0**   | **99.2**   | **98.1**   | **88.9**   | **64.4**  | **95.4**  | **85.0**   |
>
>
>
> **W4. The paper uses Llama2 as the base model. However, its multilingual performance is far from sota. Until submission, I believe Mistral, Gemma, and Llama3 have been released, whose multilingual performance is much better.**
>
> - In fact, **we have compared Mistral as well as the larger MetaMath-Llama-13B in Table 5**. The experimental results show that our method is also consistently effective on LLM with stronger multilingual capabilities.
> - We further compare our method with baselines based on Llama3 in the following table. Our method still outperforms all baselines.
>
> | Llama3, MGSM     | Bn    | Th    | Sw    | Ja    | Zh    | De    | Fr    | Ru    | Es    | En    | Lrl   | Hrl   | Avg.   |
> |------------------|-------|-------|-------|-------|-------|-------|-------|-------|-------|-------|-------|-------|--------|
> | MonoReason       | 40.4  | 53.2  | 32.0  | 53.2  | 58.0  | 64.4  | 67.2  | 67.6  | 69.2  | 76.4  | 41.9  | 65.1  | 58.2   |
> | MonoReason       | 41.2  | 55.6  | 29.2  | 50.0  | 56.8  | 62.4  | 64.4  | 66.4  | 66.0  | 74.8  | 42.0  | 63.0  | 56.7   |
> | MultiReason-Lora | 52.0  | 62.0  | 52.4  | 58.8  | 64.8  | 66.0  | 68.8  | 72.4  | **74.0**  | 75.2  | 55.5  | 68.6  | 64.6   |
> | MultiReason-SFT  | 39.2  | 47.6  | 48.0  | 48.0  | 46.4  | 52.8  | 48.0  | 47.2  | 52.8  | 57.2  | 44.9  | 50.3  | 48.7   |
> | QAlign           | 50.0  | 59.6  | 56.0  | 54.0  | 58.4  | 62.4  | 63.6  | 69.6  | 70.8  | 73.2  | 55.2  | 64.6  | 61.8   |
> | LangBridge       | 49.2  | 51.6  | 56.4  | 48.0  | 54.8  | 69.6  | 68.4  | 67.6  | 69.6  | **78.0**  | 52.4  | 65.1  | 61.3   |
> | Translate-En     | 52.0  | 41.6  | 58.0  | 54.8  | 53.2  | 63.6  | 60.4  | 55.6  | 67.6  | 76.4  | 50.5  | 61.7  | 58.3   |
> | **MergeMinds**       | **64.4**  | **65.6**  | **66.4**  | **62.0**  | **68.0**  | **71.6**  | **72.4**  | **73.2**  | 72.8  | 75.2  | **65.5**  | **70.7**  | **69.2**   |

---

> ### Author Response · Authors · 2024-08-07
> **(3/3) Response to Reviewer iF4J**
>
> **Q1. The details of the two training stages lack clarity, specifically regarding the utilization of translation and parallel data, as well as the precise training tasks involved.**
>
> - Thank you for your feedback, we will improve the description of training data in the next version of our paper.
> - In the first stage (mapping stage), all tasks including Math, Commonsense, and NLI are trained using non-English to English translation data. The following is an example:
>
> | Mapping stage (Translation data)                                                                                                           |
> |--------------------------------------------------------------------------------------------------------------------------------------------|
> | **Input (Zh):** 文档中没有元素属于该组时，该名称会被作为根结点显示在结构树上。                                                                                            |
> | **Training Target:** It will be shown in the structure tree as a top node when there are no elements belonging to this group in the document. |
>
> - In the second stage (augmentation stage), the input query of each task is translated into non-English to construct multilingual training samples, both English and non-English samples are used to train our model. Below are examples of training samples for three tasks:
>
> |             | Augmentation stage (Query translation task data)                                                                                                                                                                                                                                                                                                                                                                                                                                                                                                                                         |
> |-------------|------------------------------------------------------------------------------------------------------------------------------------------------------------------------------------------------------------------------------------------------------------------------------------------------------------------------------------------------------------------------------------------------------------------------------------------------------------------------------------------------------------------------------------------------------------------------------------------|
> | Math        | **Input (Zh):** 伯特每天都填报纸上的每日填字游戏。他每两周就用完一支铅笔。平均而言，他用完一支铅笔需要1050个字。每个填字游戏平均有多少个字？ <br> **Input (En):** Bert fills out the daily crossword puzzle in the newspaper every day. He uses a pencil to fill out the puzzles every two weeks. On average, it takes him 1050 words to use up a pencil. How many words are in each crossword puzzle on average? <br> **Training Target:** If Bert uses up a pencil to fill out the puzzles every two weeks and it takes him 1050 words to use up a pencil ... 1050/14 = 75 words in each daily crossword puzzle on average. #### The answer is: 75 |
> | Commonsense | **Input (Zh):** 酒后驾车的人可能会被指控什么? (A)惩罚 (B)逮捕 (C)车祸 (D)胡言乱语 (E)酒后驾驶 <br> **Input (En):** What is someone operating a vehicle likely to be accused of after becoming inebriated? (A) punish (B) arrest (C) automobile accidents (D) talking nonsense (E) drunk driving <br> **Training Target:** E                                                                                                                                                                                                                                                                                          |
> | NLI         | **Input (Zh):** Premise: 她不太明白。Hypothesis: 事实上，她没有理解 <br> Input (En): Premise: She doesn’t really understand. Hypothesis: Actually, she doesn’t get it. <br> **Output:** Entailment                                                                                                                                                                                                                                                                                                                                                                                                      |
>
>
>
> Thank you again for your comments. We hope our responses can resolve your concerns. We hope you can improve our score appropriately and are willing to have more discussions with you.
>
> Best wishes.

---

> ### Comment · Area_Chair_868M · 2024-08-13
>
> Dear reviewer,
> could you check at the authors new experimental results and see whether it address your concerns, and update the score accordingly if they do?

---

> ### Comment · Reviewer_iF4J · 2024-08-13
>
> Thanks so much for so detailed reply in rebuttal to address my concern. Thay addressed my concerns, and I have raised my score accordingly.

---

### Decision · Program_Chairs · 2024-09-25

**Decision:**

Accept (poster)

**Comment:**

The paper presents a method to improve multilingual capability of LLMs by providing backbone LLMs the undecided query representation from multilingual model to LLM that is more optimized for reasoning. The proposed merging recipe includes two stages (1) mapping multilingual model into the backbone LLM and (2) teaching models to perform task given mapped multilingual model’s representation and backbone LLM’s representation.

The reviewer (iF4J) have asked for more thorough evaluation against with the baselines, making input setting more equivalent and trying different base LLMs, and the proposed method consistently shows gains in these new settings. The approach is evaluated throughly on various benchmarks, including generation benchmark (suggested by reviewer q66P) and efficiency comparison (suggested by reviewer VBLh). These results strengthens the value of the proposed approach.